# Cell type-specific calcium imaging of central sensitization in mouse dorsal horn

Charles Warwick[1], Joseph Salsovic[1], Junichi Hachisuka [1,3], Kelly M. Smith[1], Tayler D. Sheahan [1], Haichao Chen[1,4], James Ibinson [2], H. Richard Koerber [1] ✉ & Sarah E. Ross [1,2] ✉

Allodynia is a state in which pain is elicited by innocuous stimuli. Capsaicin applied to the skin results in an allodynia that extends to a broad region beyond the application site. This sensitization is thought to be mediated by spinal networks; however, we do not have a clear picture of which spinal neurons mediate this phenomenon. To address this gap, we used two-photon calcium imaging of excitatory interneurons and spinal projection neurons in the mouse spinal dorsal horn. To distinguish among neuronal subtypes, we developed CICADA, a cell profiling approach to identify cell types during calcium imaging. We then identified capsaicin-responsive and capsaicin-sensitized neuronal populations. Capsaicin-sensitized neurons showed emergent responses to innocuous input and increased receptive field sizes consistent with psychophysical reports. Finally, we identified spinal output neurons that showed enhanced responses from innocuous input. These experiments provide a population-level view of central sensitization and a framework with which to model somatosensory integration in the dorsal horn.

Persistent noxious input drives changes in central circuitry that alters the integration of somatosensory stimuli, resulting in abnormally amplified pain[1]. This central sensitization can be protective in the context of acute injury because it engenders hypervigilance during recovery. In the context of chronic pain, however, central sensitization is maladaptive, leading to suffering. Given the link between central sensitization and the chronification of pain, it is important to understand activity-induced changes in the central processing of nociception.

The spinal dorsal horn processes heterogeneous primary afferent input through complex local interneuron circuits. These neural circuits contribute to spinally mediated reflexes[2] and transmit nociceptive signals to a variety of supraspinal locations by way of spinal output neurons, particularly those in lamina I, most of which collateralize within the lateral parabrachial nucleus[2–6]. The large diversity of cell types within the superficial laminae have been defined with a variety of methods including morphology, location, intrinsic firing properties,

expression profiling, and neurochemical markers[7] but we have yet to reach a consensus on the number and identity of bona fide cell types. Numerous studies have investigated the role of a given population for normal sensation and in the context of allodynia[8–17]. A common thread in many of these studies is that any disruption in the balance between inhibition and excitation can drive allodynia via loss of inhibitory tone[2,11,12,17–25] or sensitization of excitatory pathways[26–30] including spinoparabrachial (SPB) neurons[28,31–35].

Capsaicin-induced hypersensitivity is a model of central sensitization that is well characterized in psychophysical[36,37] and electrophysiological studies[38–40]. In this model, cutaneous application of capsaicin activates TRPV1[+] C-fibers resulting in burning pain that lasts minutes, and central sensitization in the dorsal horn that lasts hours. Two salient features of this central sensitization are that low threshold mechanical stimuli, which are normally innocuous, are perceived as painful (allodynia), and that the area of allodynia (the secondary zone) is much larger than the area of capsaicin-treated skin. In light of these

[1]Department of Neurobiology and the Pittsburgh Center for Pain Research, University of Pittsburgh, Pittsburgh, PA, USA. [2]Department of Anesthesiology, University of Pittsburgh, Pittsburgh, PA, USA. [3]Present address: Spinal Cord Group, Institute of Neuroscience and Psychology, University of Glasgow, Glasgow, UK. [4]Present address: School of Medicine, Tsinghua University, Beijing, China. ✉e-mail: rkoerber@pitt.edu; saross@pitt.edu

findings, capsaicin-induced allodynia is thought to be a central phenomenon, likely due to plasticity at the level of the spinal cord[39,41,42]. Consistent with this idea, electrophysiological studies have revealed that there are synaptic changes in spinal projection neurons after capsaicin treatment[38,43]. However, the identity of which spinal populations are involved in generating this plasticity have yet to be examined.

Ca$^{2+}$ imaging is a powerful approach to visualize neural activity at a population level, and several groups have applied this approach to the spinal dorsal horn[26,44–50]. However, imaging studies in this region are challenging to perform due to respiratory movement and optical access[46] and difficult to interpret due to neuronal diversity[51]. In the ex vivo somatosensory preparation, the skin, nerves, dorsal root ganglia, and spinal cord are dissected out in continuum, allowing the application of quantitative sensory stimuli to the skin. This reduced preparation is advantageous because it provides superior image quality and stability, eliminates the confound of anesthesia, and facilitates extensive pharmacological manipulation. Additionally, rather than imaging a mixed population of unidentified neurons, we focused on excitatory neurons, and further distinguished interneurons from projection neurons by virtue of retrograde labeling from the lateral parabrachial nucleus. To further differentiate the excitatory populations, we leveraged the heterogeneous expression of Gq-coupled G-protein coupled receptors (GPCRs). Because activation of Gq-coupled GPCRs gives rise to the release of Ca$^{2+}$ from internal stores, the application of GPCR ligands allows us to visualize which cells express which receptors. We have called this approach CICADA for Cell-type Identification by Ca2 + -coupled Activity through Drug Activation.

Here, we used 2-photon Ca$^{2+}$ imaging in the superficial dorsal horn to define the response properties of both spinoparabrachial neurons and excitatory spinal interneurons to quantitative mechanical and thermal stimuli. Using CICADA, we defined seven excitatory interneuron populations based on their responses to Gq-coupled GPCR agonists. We then determined how central sensitization alters their response properties and identified distinct CICADA-defined populations that comprise capsaicin-responders and capsaicin-sensitized populations. Within identified capsaicin-sensitized populations, some show selectively amplified responses to low threshold input suggestive of allodynia, whereas others show increases in receptive field size consistent with a secondary zone of allodynia expanding beyond the capsaicin injection site. Finally, we show that capsaicin-induced central sensitization alters the response properties of SPB neurons, which show a shift in tuning towards low threshold input. Together, these data represent a population view of central sensitization in the superficial dorsal horn.

## Results

### Ex vivo Ca$^{2+}$ imaging in the spinal dorsal horn

We performed two-photon Ca$^{2+}$ imaging of excitatory neurons in the superficial dorsal horn of the spinal cord in response to natural cutaneous stimulation (Fig. 1a). For these recordings, we used an ex vivo somatosensory preparation in which the spinal cord, nerves (saphenous and lateral femoral), and skin (dorsal hind paw through proximal hip) were dissected in continuum (Fig. 1b). This approach allowed us to manipulate sensory input at the skin with mechanical, thermal, and chemical stimuli. Excitatory neurons were selectively labeled using the *Vglut2-ires-Cre* allele together with Ai96, enabling Cre-dependent expression of *GCaMP6s* from the Rosa locus, and SPB neurons were identified by back-labeling from the lateral parabrachial nucleus with a vital, red-fluorescent dye, DiI (Fig. 1c). To verify that GCaMP6s was selectively expressed in excitatory neurons, we performed dual fluorescence in situ hybridization (FISH) using probes against *Gcamp6s* and *Vglut2*, and found that 97% of *GCaMP6s* neurons were positive for *Vglut2* (Fig. 1d, e). In two-photon imaging experiments, we recorded activity across neurons of different laminae by imaging from multiple

optical planes simultaneously (Fig. 1f). In this study, we focused on neurons ranging from 0 – 60 µm below the dorsal surface of the spinal cord (lamina I, IIo and IIi), but this approach allows us to visualize deeper laminae, extending down to at least 150 µm from the surface of the spinal cord (Supplementary Fig. 1). Within a 528 ×221 µm field of view, this labeling and imaging strategy enabled the visualization of ~300 excitatory neurons and ~6 to 8 SPB neurons per animal. In total, 11 mice were imaged with 2,914 individual cells recorded across all sets of experiments. Light brushing of the skin gave rise to Ca$^{2+}$ transients in a subset of neurons that was reproducible (Fig. 1g, h) and showed no significant rundown of response after repeated stimulation (Supplementary Fig. 2).

To characterize the functional response properties of excitatory neurons in the dorsal horn, we applied a series of mechanical and thermal stimuli to the skin. For static mechanical stimulation, we used von Frey filaments of 2.0 g for high threshold (HT) or 0.16 g for low threshold (LT). For dynamic mechanical stimulation, we applied a wide brush or an air puff. Thermal stimuli consisted of fast and slow ramps to warm (40 °C), hot (52 °C), cool (18 °C), or cold (4 °C) temperatures using a Peltier thermode. Examples of the Ca$^{2+}$ responses in several excitatory interneurons and SPBs are shown (Fig. 1i), illustrating that individual neurons have complex responses to diverse sensory inputs. We found that ~70% of GCaMP6s-expressing neurons within the field of view responded to one or more of these natural stimuli.

### Polymodality and tuning

As a first step to functionally classify neurons in the dorsal horn, we defined a responder as a cell showing a stimulus-evoked Ca$^{2+}$ transient with a peak amplitude that was at least 125% ΔF/F and more than 6 standard deviations above baseline activity (Fig. 2a and Supplementary Fig. 3). Of cells that responded to cutaneous stimuli, we found that 86% of excitatory interneurons responded to mechanical, 58% to heat, and 28% to cold (Fig. 2b). Among SPB neurons these numbers were even higher: 100% responded to mechanical, 52% to heat, and 47% to cold. Indeed, compared to interneurons, the fraction of SPB neurons that responded to all three cardinal modalities (mechanical, heat, and cold) was significantly higher, indicating a greater degree of modality convergence (Fig. 2c). Across mechanical sub-modalities, we observed that HT (2.0 g), LT (0.16 g), and light brushing stimuli activated 82%, 59% and 37% of superficial interneurons, and 100%, 52% and 53% of SPB neurons, respectively (Fig. 2d). Again, the proportion of SPB neurons that responded to all three mechanical stimuli (pan-mechanical) was significantly higher than that of interneurons (Fig. 2e). Thus, many spinal interneurons and even more SPB neurons show polymodal responses to cardinal sensory modalities and respond broadly to a variety of subtypes of mechanical stimuli.

In light of this widespread polymodality, we speculated that additional information might be encoded within the magnitude of response to each modality and/or sub-modality. We thus developed a two-dimensional vector-based method to visualize and quantify the tuning of SDH cells (Fig. 2f). For each cell, three vectors were calculated based on the amplitude of the cell's responses to mechanical, heat, and cold stimuli, with vectors for each modality plotted at 120° angles from one another. The overall tuning of a cell was determined by summing these three vectors, giving rise to a single line in which the direction indicates the relative modality preference and the length indicates the magnitude of preference. Applying this vector-based tuning, we found that although many cells are polymodal they nonetheless show tuning preferences across cardinal modalities (Fig. 2g) and mechanical sub-modalities (Fig. 2h).

### Somatotopy and functional organization

Cells within the dorsal horn are organized somatotopically, based on in vivo single unit recordings[52–54], but how this organization is manifest at a population level has never been functionally visualized. To

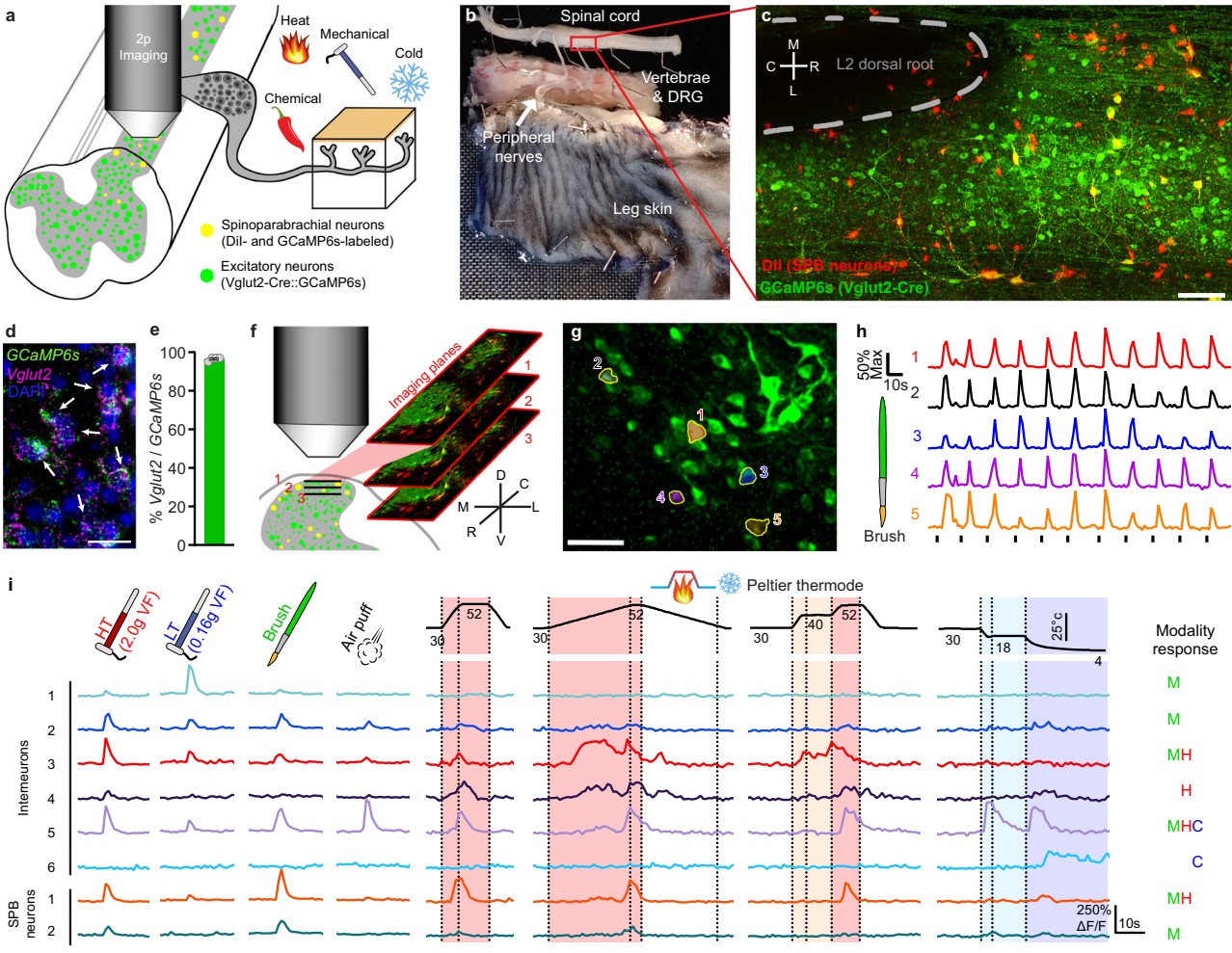

**Fig. 1 | Population imaging in the ex vivo somatosensory preparation.**
**a** Schematic of the ex vivo somatosensory preparation and neuronal labeling strategy. **b** Picture of the ex vivo somatosensory preparation with a typical recording region rostral of the L1-L2 dorsal root entry zone highlighted in red. **c** Low-magnification fluorescence image of the L2 dorsal root entry zone (scale bar, 100 μm) showing GCaMP6s expression in *Vglut2-Cre* positive excitatory neurons (green) and SPB neurons (red). *N* = 11 animals. Rostral (R), caudal (C), medial (M), and lateral (L) are indicated on the white cross. **d** Representative fluorescence in situ hybridization of *Gcamp6s* (green) and *Vglut2* (magenta) with DAPI (blue) in the superficial dorsal horn (scale bar, 25 μm). **e** Percent of *GCaMP6s* SDH neurons which express *Vglut2* (*n* = 4 mice, 60–76 *GCaMP6s* + nuclei quantified/mouse). Data are presented as mean values ± SEM. **f** Schematic of the multi-plane (XYZT) imaging protocol used for Ca²⁺ imaging. 3 separate optical planes with a spacing of 14 μm are

imaged simultaneously allowing parallel access to lamina I and II neurons. Rostral (R), caudal (C), dorsal (D), ventral (V), medial (M), and lateral (L) are indicated for the labeled image planes. **g** Example maximum ΔF/F image with 5 brush responsive cells circled (scale bar, 50 μm). *N* = 11 animals. **h** ΔF/F Ca²⁺ traces from the 5 cells circled in **g** showing consistent responses to repeated brushing of the skin. **i** Top: stimulation type for mechanical and thermal stimuli. High threshold (HT) and low threshold (LT) stimuli were applied serially across an area of skin 15 × 15 mm, while all other stimuli covered the entire area with each trial. Shaded region represents the period during which responsiveness is considered. Bottom: representative ΔF/F Ca²⁺ traces from both interneurons and SPB neurons in response to the indicated cutaneous stimuli. Responsiveness to cooling (C, blue) mechanical stimulation (M, green) or heating (H, red) is indicated on the right.

examine this feature, we probed the skin with a LT filament at proximal, intermediate, or distal regions (Fig. 3a). This receptive field mapping showed that the stimulation of proximal skin activated neurons that are more lateral within the dorsal horn, whereas stimulation of a distal skin activated neurons that are more medial (Fig. 3b, c). These data are consistent with prior studies indicating that the spinal cord is somatotopically organized such that inputs pertaining to a given area of skin are mapped across the medio-lateral and rostro-caudal axis in partially overlapping, elongated rostro-caudally oriented ellipses[52–56] with modality and intensity information for each area of input contained within a column oriented in the dorsal-ventral axis (Supplementary Fig. 4). Next, we examined how the intensity of stimulation affected receptive field size by stimulating at 16 sites equally spaced in a 4 × 4 grid using a HT, LT, or airpuff stimulus and summing the number of responsive sites for each stimulation, averaged across 2 trials (Fig. 3d). We observed that the receptive field size of individual

excitatory neurons was significantly affected by the stimulus intensity: the determined receptive field size with a HT filament was significantly greater than that with a LT filament, which was significantly greater than that with airpuff (Fig. 3e, f). We also found that neurons that were closer to the surface (*i.e.*, lamina I) had significantly larger receptive fields than neurons that were deeper (*i.e.*, lamina II) (Fig. 3g–i).

## CICADA: Cell-type identification by Ca²⁺-coupled activity through drug activity
A major limitation of typical population imaging studies is that, although one can visualize Ca²⁺ transients in many neurons simultaneously, it is difficult to determine the specific identity of these neurons, thereby hampering the degree to which the data can be interpreted. To overcome this limitation, we developed a cell profiling strategy that leverages the fact that different subtypes of neurons express different combinations of GPCRs. We reasoned that

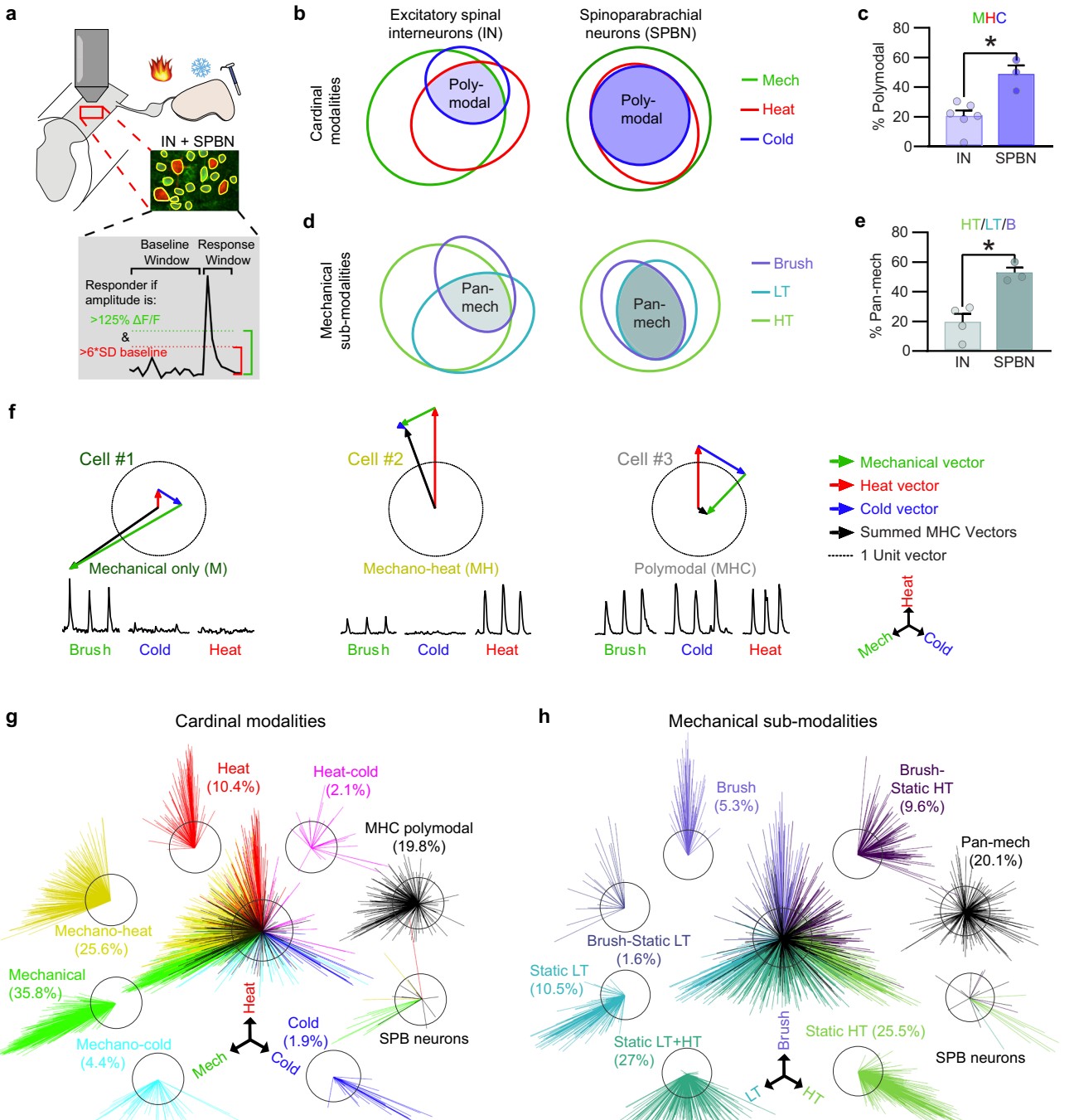

**Fig. 2 | Polymodality and tuning among excitatory interneurons and projection neurons. a** Experimental design of imaging (top) and schematic of response thresholds (bottom). **b** Proportions of excitatory interneurons (IN, left) and SPB neurons (SPBN, right) that were responsive to cardinal modalities (cold, mechanical, heat). Shaded region represents the MHC polymodal population corresponding to **c**. **c** Percent of INs and SPBNs (averaged by animal) which displayed cardinal polymodality. *$P < 0.05$ (two-tailed unpaired t test, $P = 0.0051$). $N = 6$ animals for interneurons $N = 3$ animals for SPBNs. Data are presented as mean values ± SEM. **d**, Proportions of INs (left) and SBPNs (right) that were responsive to the mechanical sub-modalities: brush (purple), static low threshold (LT, blue), or static high threshold (HT, green). Pan-mechanical sensitivity (pan-mech) indicates responsiveness to all tested types of mechanical stimuli. Shaded

region represents the MHC polymodal population corresponding to **e**. **e** Percent of INs and SPBNs (averaged by animal) which were pan-mechanical. *$P < 0.05$ (two-tailed unpaired t test, $P = 0.0068$). $N = 4$ animals for interneurons and $N = 3$ animals SPBNs. Data are presented as mean values ± SEM. **f** Schematic and examples of vector-based tuning. Top: ΔF/F Ca$^{2+}$ traces evoked by brush (green), cold (blue), and heat (red). Bottom: modality response vectors (colored) and summed vectors indicating the final tuning (black). **g, h** Cardinal **g** and mechanical sub-modality **h** tuning of SDH interneurons grouped by their threshold-based categorization. SPBNs are shown separately. Black circle is 1 unit vector in diameter. $N = 1265$ cells for the cardinal modalities and 1080 cells for the mechanical sub-modalities from 5 mice including 22 SPB neurons and 6 mice for interneurons.

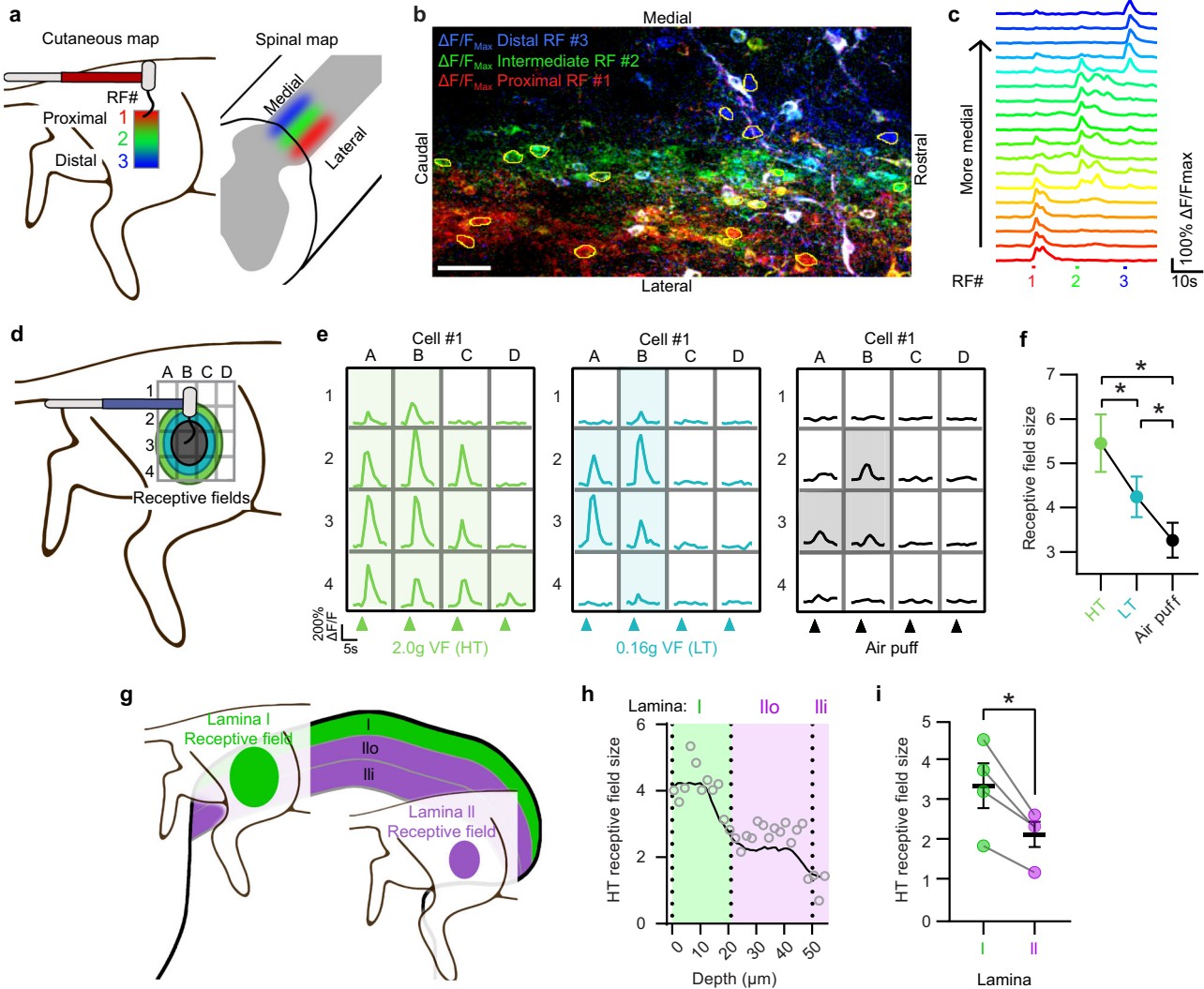

**Fig. 3 | Receptive fields in SDH are shaped by somatotopy, modality, and relative dorso-ventral position. a** Schematic illustrating proximal-distal somatotopy. **b** Field of view in the superficial SDH showing maximum intensity responses after LT stimulation pseudo-colored to each of the indicated receptive fields (RF) in **a**: red, proximal; green, intermediate; blue, distal (scale bar, 50 μm). N = 4 animals. **c**, ΔF/F Ca²⁺ traces from circled cells in **b**, showing responses to stimulation at the three indicated RFs, serially stimulated from proximal to distal. Traces are arranged by their somatic medio-lateral location. **d** Schematic illustrating RF sizes for different mechanical stimuli: HT, green; LT, blue; and air puff, black. **e** Representative ΔF/F Ca²⁺ traces at 16 RFs for each stimulation type. **f** Mean ± SEM receptive field size of excitatory SDH neurons responsive to all 3 modalities was calculated by summing the total number of responses observed at each of the sites. There was a significant main effect of stimulation type ($F$ (1.686, 227.6) = 49.3), $P = <0.0001$ (1-way RM ANOVA). Post hoc testing was performed comparing all stimulation types. N = 136 cells; *$Q = <0.0001$. **g** Schematic illustrating dorso-ventral differences in RF size, between lamina I (green) and lamina II (purple). **h** HT RF size plotted as a function of dorso-ventral depth. **i** Mean ± SEM RF size of neurons located within lamina I and lamina II. *$P < 0.05$ (two-tailed paired t test, $P = 0.0220$). N = 4 mice.

pharmacological profiling could be used to visualize which cells express which receptors because activation of Gq-coupled GPCRs gives rise to IP3-mediated release of Ca²⁺ from internal stores. To isolate the direct effects (cell autonomous) from indirect effects (network responses) we included the Na⁺ channel blocker tetrodotoxin (TTX) in the bath solution, which prevents action potential propagation of neuronal activity. In the presence of TTX, the cells that show a Ca²⁺ transient in response to a given agonist are only those that express its receptor, allowing us to unambiguously define cell types based on their responses to GPCR agonists applied in series at the end of each experiment (Fig. 4a).

To determine which GPCR agonists would be useful for CICADA profiling, we screened 15 agonists for consistent responses in superficial excitatory neurons. This initial list represented the majority of Gq-coupled receptors with known ligands that are well expressed in the dorsal horn, based on RNAseq studies[13]. When applied in random order, eight of these ligands gave rise to robust

Ca²⁺ transients in the superficial dorsal horn: oxotremorine M (Oxo.), an agonist for the muscarinic acetylcholine receptor M3 (CHRM3); gastrin-releasing peptide (GRP), an agonist of the gastrin-releasing peptide receptor (GRPR); taltirelin, an agonist of the thyrotropin-releasing hormone receptor (TRHR); neuromedin-B (NMB), an agonist of the neuromedin-B receptor (NMBR); cholecystokinin (CCK), an agonist of the cholecystokinin type B receptor (CCKBR); Substance-P (SP), an agonist at the Substance-P receptor (TACR1); oxytocin (OXT), an agonist at the oxytocin receptor (OXTR); and Neurokinin-B (NKB), an agonist at the tachykinin receptor 3 (TACR3) (Fig. 4b, Supplementary Fig. 5). Overall, 25% of excitatory neurons responded to one or more CICADA ligands, although this proportion varied across laminae (32% in lamina I, 70% in lamina IIi, and 15% in lamina IIo; Supplementary Fig. 5f). Because the combinatorial responses to these ligands had the potential to be complex (Supplementary Fig. 5c–e), we next performed dimensionality reduction using K-means clustering[57]. This analysis yielded seven groups of

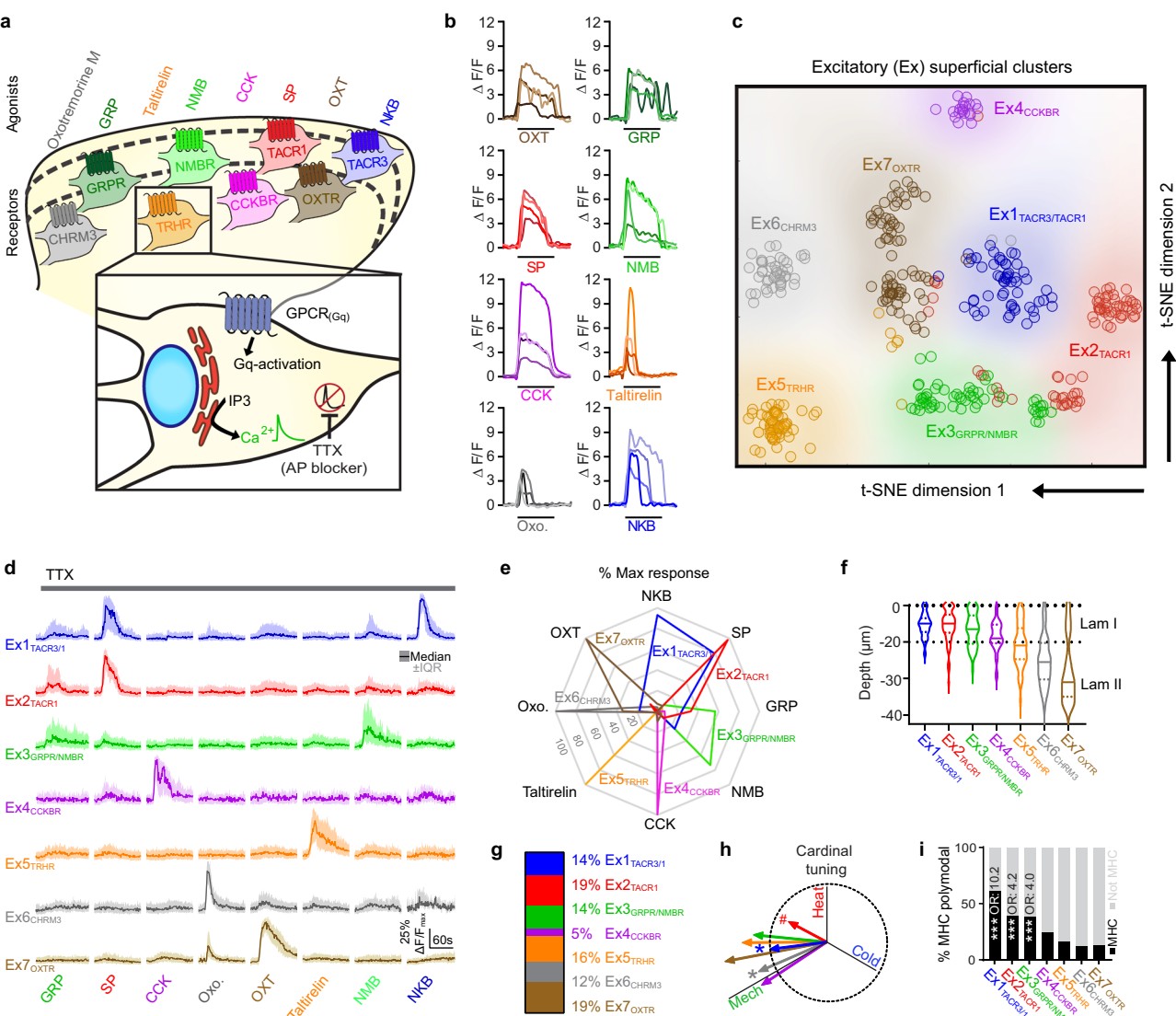

**Fig. 4 | CICADA: Cell-type identification by Ca²⁺ coupled activity through drug activity. a** Overview of CICADA showing the intracellular mechanisms (inset) and the agonists used to target the specified receptors (top). **b** Representative ΔF/F Ca²⁺ traces for each of the CICADA ligands. Each trace is a different cell. SP, 1 μM; OXT, 1 μM; GRP, 300 nM; Oxotremorine M (Oxo.) 50 μM; Taltirelin, 3 μM; NMB, 100 nM; CCK, 200 nM; NKB, 500 nM. **c** t-SNE plot of excitatory superficial clusters (Ex) identified by CICADA. Each dot represents a CICADA-identified cell, color-coded for the appropriate cluster with the cluster regions color coded by shading. Cluster numbers (*e.g.* Ex2) are followed by the principal receptor found in that population of cells (Ex2$_{TACR1}$). **d** Normalized Ca²⁺ traces of each CICADA ligand for Ex1-7. Each cell's ΔF/F trace was normalized to a maximum of 1 and minimum of 0. Data are shown as median ± interquartile range. Ex1$_{TACR3/1}$, 57 cells; Ex2$_{TACR1}$, 75 cells; Ex3$_{GRPR/NMBR}$, 57 cells; Ex4$_{CCKBR}$, 21 cells; Ex5$_{TRHR}$, 65 cells; Ex6$_{CHRM3}$, 49 cells; Ex7$_{OXTR}$, 74 cells (cells are pooled from 4 mice). **e** Radar chart showing the response to CICADA ligands within each population. The spoke

length is the population average of normalized response to CICADA ligands (($X$-$X_{min}$)/$X_{max}$). **f** Violin plot showing the depth of each population relative to the surface of the dorsal grey matter. **g** Relative abundance of neurons making up the excitatory populations identified by CICADA. **h** Cardinal tuning of Ex1-7. For vector angles: 1-way ANOVA ($F$ (7, 1541) = 3.983), $P$ = 0.0003. For vector magnitudes: 1-way ANOVA ($F$ (7, 1541) = 3.289), $P$ = 0.0018. Post hoc testing was performed comparing each population's vector angle (#$Q$ < 0.05) and magnitude (*$Q$ < .05) to the average of all recorded cells. Dotted circle is 1 unit vector in diameter. **i** Percent of cells within each cluster which show cardinal (MHC) polymodal. Multiple logistic regression model was used to test how predictive each cluster was for MHC modality, ***$P$ = .0001 ($P$ values are calculated from an F test). Odds ratio (OR) shown for significant results. Full logistic regression model shown in source data. As described in methods: all tests were 2 tailed and any post hoc comparisons were corrected for multiple comparisons by False Discovery Rate correction.

excitatory neurons (Fig. 4c). The cluster-averaged Ca²⁺ traces revealed that most clusters were defined by a high amplitude response to one or two CICADA ligands (Fig. 4d, e). Moreover, cell clusters showed distinct distributions according to depth, consistent with the idea that they represent populations of bona fide cell types (Fig. 4f, Supplementary Table 1). We therefore labeled these populations Ex1 through Ex7 according to their relative location along the dorso-ventral axis and added subscript labels indicating cognate receptors. We found that the relative frequency of each population was broadly similar (~15%), with Ex4$_{CCKBR}$ being noticeable rarer (5%)

(Fig. 4g). Notably, populations were significantly different from one another with respect to their tuning and response properties (Fig. 4h, Supplementary Fig. 6a–c), with populations in lamina I showing a greater degree of polymodality (Fig. 4i) than deeper populations. We also found that the tuning of lamina I CICADA populations were distinct from depth matched cells which were not marked by CICADA (Supplementary Fig. 6d–g). This suggests that even when the relative laminar location is controlled for, CICADA marks distinct populations. Thus, CICADA is a simple strategy to define populations of putative cell types in dorsal horn Ca²⁺ imaging experiments.

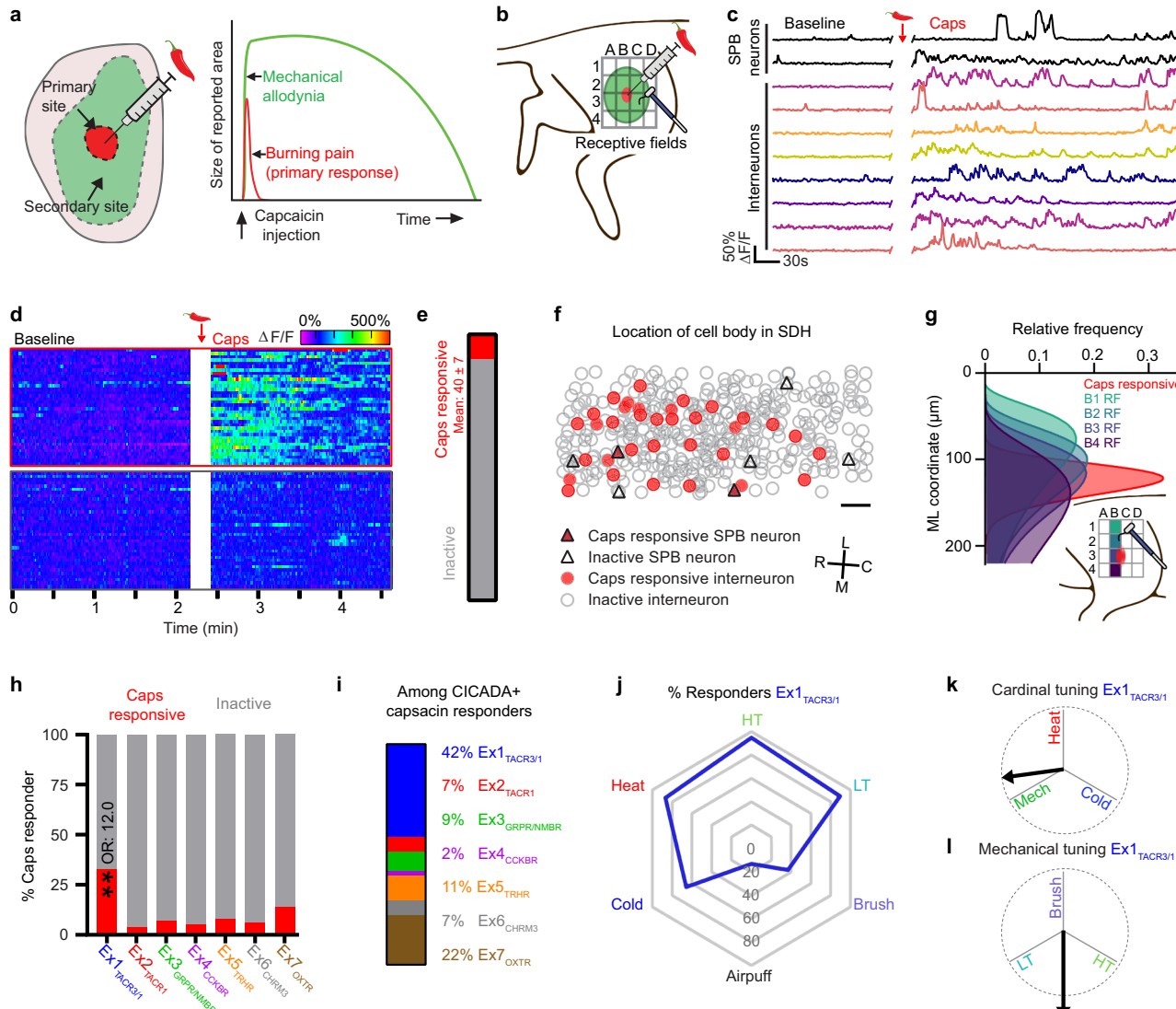

**Fig. 5 | Intradermal capsaicin rapidly activates a localized subset of excitatory cells which are enriched within Ex1TACR3/1. a** Schematic of intradermal capsaicin (caps) model of central sensitization. Relative size and location of the primary and secondary sites are shown on the left and duration/intensity of the psychophysical phenomena within each region are shown on the right. **b** Schematic for capsaicin injection and testing in the ex vivo preparation, the grey grid shows the testing sites for receptive field mapping. **c** Example $\Delta F/F$ Ca²⁺ traces from cells which were responsive to capsaicin injection. **d** Heat map of $\Delta F/F$ Ca²⁺ traces from cells within the same spinal cord which were activated or quiescent after capsaicin injection. **e** Percent of excitatory cells activated by capsaicin. N = 4 mice, mean ± SEM. **f** Somatic location in the SDH of SPB neurons (triangles) and interneurons (circles) which were either quiescent (open) or active (red) after capsaicin injection (scale bar, 50 µm). Rostral (R), caudal (C), medial (M), and lateral (L) are indicated on the black cross. **g** Using the relative distribution of cells along the medio-lateral (ML) axis activated by HT stimulation at different receptive fields (RF) we can infer the

position and proximal-distal spread of the capsaicin injection by comparing the distribution of capsaicin responders (red) to the receptive field stimulations.
**h** Percent of cells which responded to capsaicin injection within each excitatory population. A multiple logistic regression model was used to test how predictive each cluster was for being capsaicin responsive **\*\*P < .001. Odds ratio (OR) shown for significant results (Ex1TACR3/1, P = 0.0001, P values are calculated from an F test). Full logistic regression model shown in source data. **i** Relative frequency of Ex1-7 within all CICADA identified cells responsive to capsaicin. **j** Radar chart showing the percent of cells in Ex1TACR3/1 which responded to the indicated cutaneous stimuli. **k** Cardinal tuning for Ex1TACR3/1. **l** Mechanical sub-modality tuning for Ex1TACR3/1. Dotted circle is 1 unit vector in diameter. As described in methods: all tests were 2 tailed and where multiple post hoc comparisons were indicated by the main effect, a correction for multiple comparisons was implemented using a False Discovery Rate of 0.05 via the two-stage step-up method of Benjamini, Krieger and Yekutieli. Reported Q-values are the FDR corrected P-values.

## Immediate response to intradermal capsaicin

Previous human psychophysical experiments have shown that intradermal injection of capsaicin causes an immediate response consisting of brief burning pain (<5 min) that is restricted to the capsaicin injection site, followed by a period of sensitization which manifests as long-lasting mechanical allodynia (several hours) extending over a large region outside the capsaicin injection site (Fig. 5a)[37]. The mechanical allodynia in this model is understood to be the result of central, rather than peripheral, sensitization because the primary afferents that innervate the secondary zone are unchanged after the capsaicin

injection[39,41,42]. However, the specific spinal neurons and circuits involved in mediating the response has not yet been visualized at a population level.

We therefore sought to visualize how capsaicin-mediated central sensitization is manifest in the dorsal horn through Ca²⁺ imaging coupled with post-hoc cell identification through CICADA (Fig. 5b). Intradermal injection of capsaicin (7.5 µg) elicited significant increases in Ca²⁺ activity in ~40 excitatory neurons per animal, which peaked within ~2 minutes and persisted for ~5 minutes (Fig. 5c–e). These capsaicin responders, which included SPB neurons, were restricted to a

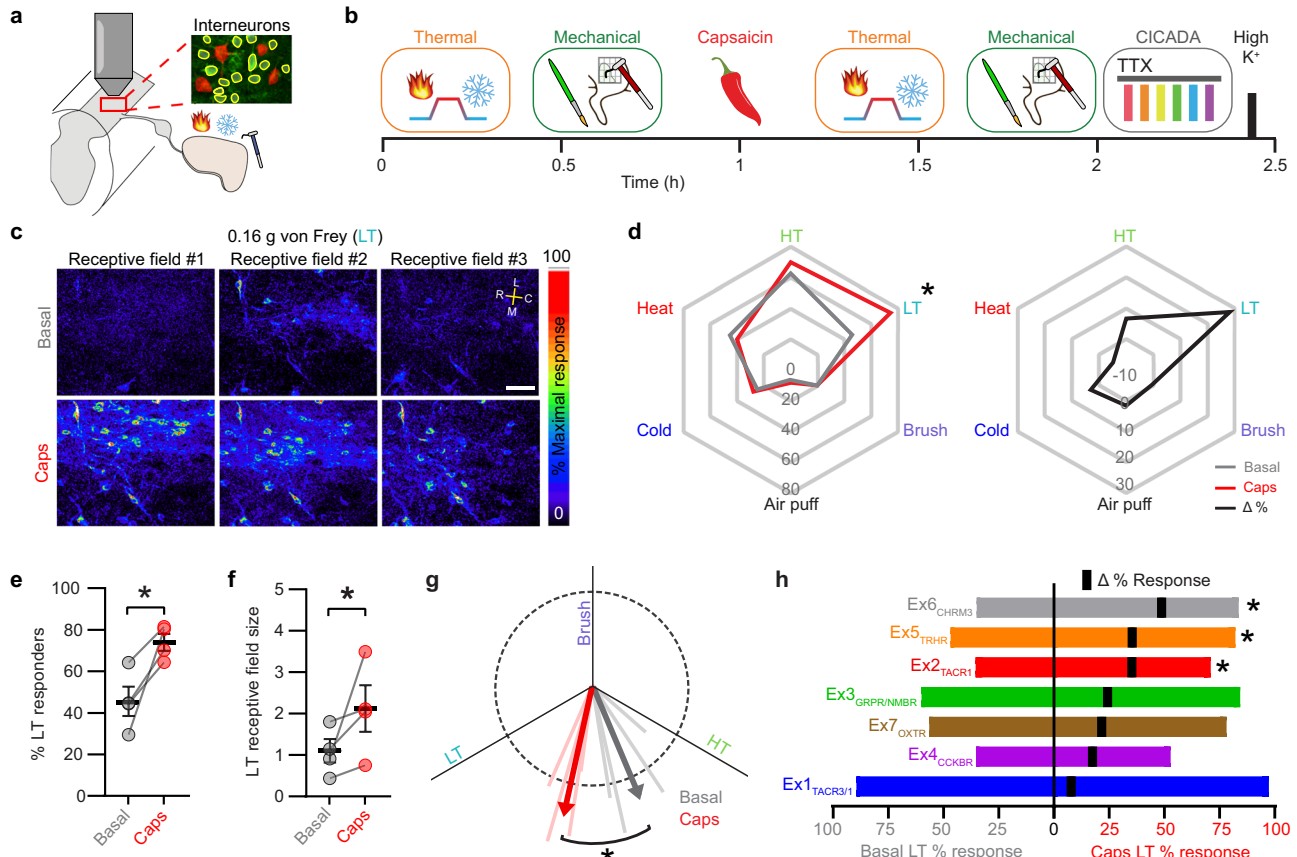

**Fig. 6 | Capsaicin injection increases the frequency of LT responsive neurons and enhances LT tuning. a** Experimental design. **b** Experimental protocol for testing the effects of capsaicin induced central sensitization. **c** Field of view in the superficial SDH showing maximum ΔF/F responses during LT stimulation at 3 RFs before and after intradermal injection of capsaicin (scale bar, 50 μm). *N* = 4 animals. **d** Left: radar chart showing the percent of excitatory cells responsive to the indicated stimuli before (grey) and after capsaicin (red). Right: change in percent responders. There was a significant main effect of stimulation (*F* (4, 12) = 41.11), *P* < 0.0001, and a significant interaction between stimulation and time (*F* (4,12) = 5.176), *P* = 0.0117 (2-way RM ANOVA). Post hoc testing was performed comparing post/pre capsaicin for all stimulations. *N* = 4 mice, **Q* < 0.05 **e** Percent of all excitatory cells responsive to LT stimulation before/after capsaicin. N = 4 animals. **Q* < 0.05. Data are presented as mean values ± SEM. **f** LT RF size of excitatory cells before/after capsaicin. Corrections for both LT and HT were included (HT data shown in Supplementary Fig. 8). There was a significant interaction between stimulation and time (*F* (1,3) = 13.34), *P* = 0.0354

(2-way RM ANOVA). Post hoc testing was performed comparing RF size before/after capsaicin. N = 4 mice, **Q* < 0.05. Data are presented as mean values ± SEM. **g**, Mechanical sub-modality tuning for excitatory cells before/after capsaicin. Mean value shown as thick arrows and individual animals shown as light, thin lines. Dotted circle is 1 unit vector in diameter. **P* > 0.05 (two-tailed paired t test, *P* = 0.0281). **h** Percent responders to LT stimuli before (left) and after (right) capsaicin for each CICADA population. The change in % responders (Δ) is indicated with a black tick mark. There was a significant main effect of capsaicin (*F* (1, 3) = 11.02), *P* = 0.0451 (Mixed-effects restricted maximum likelihood model). Post hoc testing was performed comparing post/pre capsaicin for all populations. *N* = 4 mice, **Q* < 0.05. Ex2, Q = 0.0161; Ex5 Q = 0.0157; Ex6, Q = 0.0155. As described in methods: all tests were 2 tailed and where multiple post hoc comparisons were indicated, a correction for multiple comparisons was implemented using a False Discovery Rate of 0.05 via the two-stage step-up method of Benjamini, Krieger and Yekutieli. Reported *Q*-values are the FDR corrected *P*-values.

narrow medio-lateral band with appropriate somatotopy in the dorsal horn (Fig. 5f, g, Supplementary Fig. 7). We next analyzed the CICADA-defined populations to determine which were preferentially activated by the capsaicin injection. Using multiple logistic regression, we found that Ex1TACR3/1 was the only population that showed statistically significant predictive power for capsaicin response (Fig. 5h). Among capsaicin-responsive cells within Ex1- Ex7, 42% were contained within Ex1TACR3/1 (Fig. 5i). Of note, Ex1TACR3/1 was significantly enriched in capsaicin responders relative to other lamina I cells (44% of Ex1TACR3/1 vs 8% of depth-matched excitatory neurons that were not labeled by CICADA ligands) even though Ex1TACR3/1 makes up <10% of excitatory neurons in lamina I. Thus, intradermal capsaicin drives activity in excitatory neurons in the superficial dorsal horn, and this activity is significantly enriched within Ex1TACR3/1, a population of relatively untuned (Fig. 5k, l), polymodal neurons (Fig. 5j) that are located in lamina I.

## Capsaicin-induced central sensitization
Capsaicin-induced sensitization gives rise to mechanical allodynia in which LT stimuli are perceived as noxious. To interrogate the neural manifestations of allodynia at the level of the superficial dorsal horn, we compared the responses to cutaneous stimulation before and after capsaicin treatment (Fig. 6a, b). Consistent with psychophysical studies[36,37], we found that capsaicin injection altered LT responses in 56% (+/− 6% SEM) of excitatory superficial cells without significantly affecting the responses to other modalities (Fig. 6c, d, Supplementary Fig. 8a, b). This change was manifest as both an increase in the proportion of neurons that respond to LT stimuli (Fig. 6e), the amplitude of response (Supplementary Fig. 8c–e), as well as increases in the receptive field size to LT stimulation (Fig. 6f, Supplementary Fig. 8f–i). Finally, analysis of mechanical sub-modality tuning revealed that, on average, neurons showed a significant shift in preference toward LT stimuli (Fig. 6g, Supplementary Fig. 8j–l). This finding indicates that an

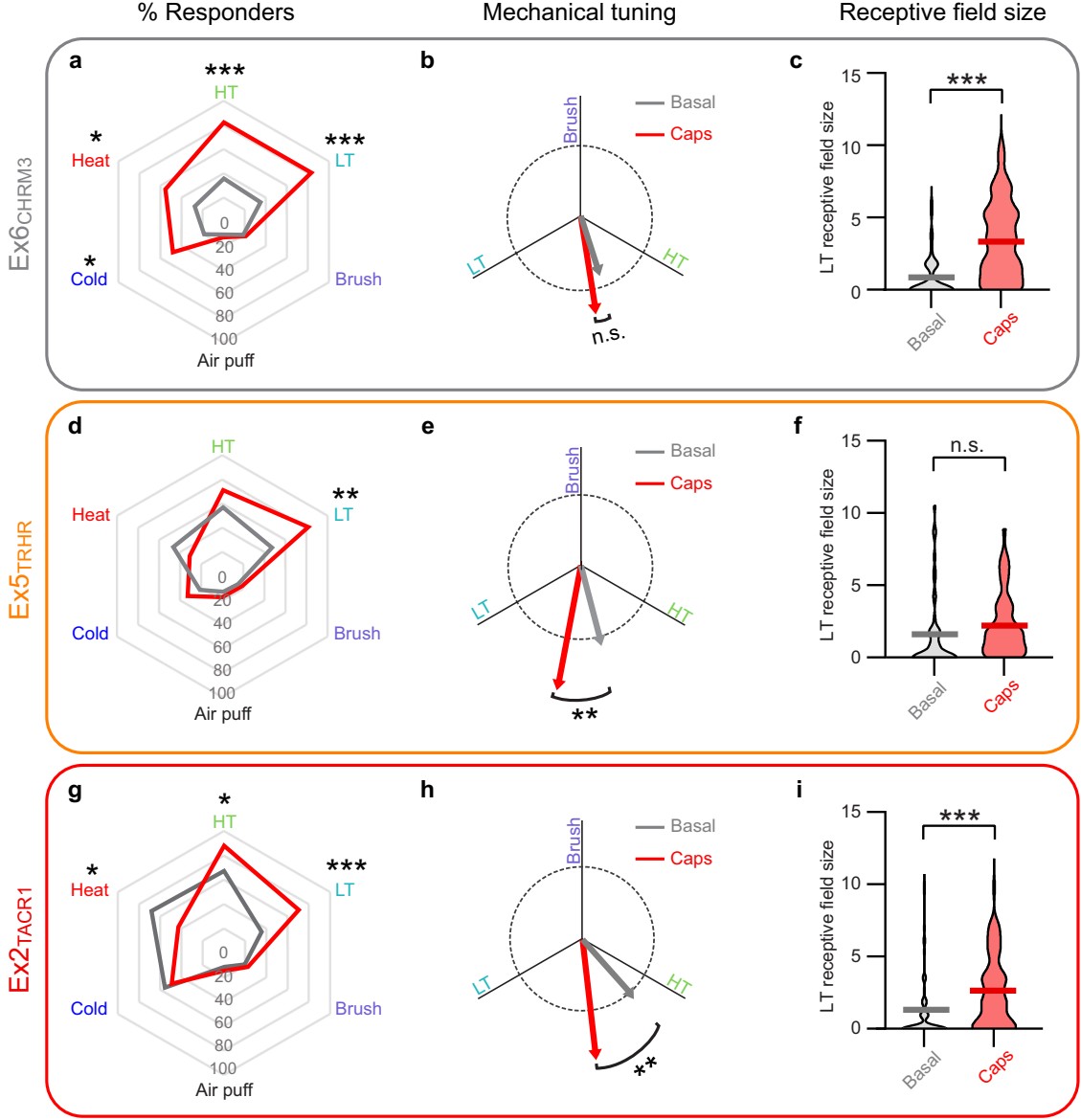

**Fig. 7 | CICADA reveals distinct populations involved in central sensitization.** Radar charts showing the percent of cells in Ex6_CHRM3 **a**, Ex5_TRHR **d**, and Ex2_TACR1 **g** which responded to the indicated cutaneous stimuli before (grey) and after capsaicin injection (red). *Q < 0.05, **Q < 0.01, ***Q < 0.001, for each population a Fisher's exact test (# responses before and after capsaicin) with FDR correction was performed for each modality, only significant results are indicated (Benjamini and Hochberg, 6 tests, exact Q values for all tests in source data). Mechanical sub-modality tuning for Ex6_CHRM3 (**b**), Ex5_TRHR (**e**), and Ex2_TACR1 (**h**) before (grey) and after capsaicin injection (red). Dotted circle is 1 unit vector in diameter. There was a significant main effect of capsaicin (F (1, 123) = 30.10),

P < 0.0001 (2-way RM-ANOVA). Post hoc testing was performed comparing post vs pre capsaicin for each group, **Q < 0.01. LT receptive field size of Ex6_CHRM3 **c**, Ex5_TRHR **f**, and Ex2_TACR1 **i**) before (grey) and after capsaicin injection (red). There was a significant main effect of capsaicin (F (1, 148) = 45.59), P < 0.0001, and a significant interaction between capsaicin and CICADA population (F (2, 148) = 5.765), P = 0.0039 (2-way RM-ANOVA). Post hoc testing was performed comparing post/pre capsaicin for all stimulations, ***Q < 0.0001. Only cells with cutaneous input were included in these analyses. Ex2_TACR1, 62 cells; Ex5_TRHR, 45 cells; Ex6_CHRM3, 43 cells (cells pooled from 4 mice). Data shown are mean of pooled cells.

emergent property following capsaicin treatment is that excitatory neurons in the superficial dorsal horn show a selective increase in the magnitude of their response to LT stimuli at a population level. To examine which cell types might be driving these capsaicin-induced changes, we compared the proportion of LT responders across CICADA-defined populations (Fig. 6h). In this analysis, three populations showed a significant increase in the proportion of cells that responded to LT stimuli post capsaicin, and were thus examined in more detail: Ex2_TACR1, Ex5_TRHR, and Ex6_CHRM3.

The largest capsaicin-induced change was observed in Ex6_CHRM3 (Fig. 7a–c). These lamina II cells rarely showed activity in response to any natural stimulus in the basal state and, of those that did respond,

showed minimal tuning preference as indicated by the small vector magnitude. After injection of capsaicin, however, Ex6_CHRM3 cells showed significantly increased responsivity across almost all modalities, including LT, HT, heat, and cold (Fig. 7a). In addition, capsaicin treatment caused a 3.9-fold increase in the size Ex6_CHRM3 receptive fields. However, the tuning of Ex6_CHRM3 remained relatively unchanged as revealed by the absence of a change in the vector angle. These findings suggest that capsaicin treatment resulted in a generalized amplification of responses within Ex6_CHRM3, independent of stimulus modality (Fig. 7b, c). Thus, capsaicin-induced central sensitization alters the response properties of neurons in the Ex6_CHRM3 population such that they increase the gain of sensory input. In addition, the

increased receptive field size to LT input suggests that Ex6$_{CHRM3}$ cells might contribute to psychophysical phenomenon that sensitization occurs beyond the original site of injury.

The second CICADA population that showed capsaicin-induced changes to LT input was Ex5$_{TRHR}$, another broadly tuned population found in lamina II that showed modest responses to natural stimuli in the basal state. Upon capsaicin-induced sensitization, Ex5$_{TRHR}$ cells showed a dramatic and selective increase in their response to LT input, with no change in their response to HT, heat, or cold (Fig. 7d). This selective increase in response to LT stimuli was manifest as a significant shift in tuning toward LT, but no significant change in receptive field size (Fig. 7e, f). These findings suggest that an emergent property of Ex5$_{TRHR}$ following capsaicin treatment is that LT stimuli begin to produce responses that are similar in magnitude to those produced by HT stimuli. Thus, Ex5$_{TRHR}$ cells are well positioned to contribute to the psychophysical phenomenon of allodynia.

The third population that showed capsaicin-induced changes was Ex2$_{TACR1}$. These neurons are a lamina I population that, under basal conditions, showed an equal preference for heat and HT mechanical stimuli (Supplementary Fig. 9), suggesting that these neurons are tuned for noxious stimuli. Following capsaicin treatment, however, Ex2$_{TACR1}$ showed a significant change in tuning towards LT input, indicating an amplification of LT stimuli (Fig. 7h) as well an increase in the percent of cells responsive to LT stimuli and an intriguing reduction in heat responsiveness (Fig. 7g). These findings raised the possibility that following capsaicin treatment, the LT-induced activity in these cells may signal nociception, and thus represent a neural correlate of allodynia. In addition, capsaicin treatment significantly increased the receptive field size of Ex2$_{TACR1}$ neurons (Fig. 7i). Because the capsaicin-induced changes observed in Ex2$_{TACR1}$ neurons appear to be a combination of the changes observed in Ex5$_{TRHR}$ and Ex6$_{CHRM3}$, these findings raise the possibility that Ex2$_{TACR1}$ could be involved in integrating multiple populations to mediate allodynia within the secondary zone.

## Allodynia is manifested in spinal output neurons

In order for the sensitization we observed in excitatory interneurons to drive the psychophysical phenomenon of allodynia, spinal projection neurons must relay these nociceptive signals supraspinally. We found that capsaicin treatment shifted the overall tuning of SPB neurons towards LT input, consistent with the effects of central sensitization producing allodynia (Fig. 8a, b). Because SPB neurons are heterogeneous with respect to their response properties, we classified them as either wide-dynamic range (WDR), high threshold (HT), or low threshold (LT) (Fig. 8c) using previously published criteria[38]. We found that both HT and WDR SPB showed significant increases in tuning to LT input, however the degree of change was much greater for WDR compared to HT SPB neurons (Fig. 8d). Finally, examining CICADA responses across populations, we found that WDR cells were more frequently responsive to CICADA agonists (71.4%), compared to either LT (40%) or HT (0%) SPB neurons. In particular, all SP-responsive SPB neurons were WDR cells (Fig. 8e). Since capsaicin is known to activate peptidergic afferents that release SP, these findings suggest that capsaicin-induced allodynia may be conveyed to the brain through a population of TACR1 expressing WDR SPB neurons and, to a lesser extent, some HT SPB neurons (Fig. 8f).

## Discussion

Here, we provide the first comprehensive functional characterization of excitatory neurons in the dorsal spinal cord at a population level. First we applied quantitative natural stimuli to the skin to determine response properties of individual neurons. To further classify these neurons, we developed CICADA, a cell profiling method to identify neuronal subsets based on their responses to Gq-coupled GPCR

ligands. We then describe the spinal representations of the intradermal injection of capsaicin, which gives rise to transient burning pain and persistent mechanical allodynia in humans. Our findings suggest that different CICADA-defined populations contribute to distinct aspects of the capsaicin response. In particular, Ex1$_{TACR3/1}$ neurons are immediately responsive to capsaicin injection, whereas Ex2$_{TACR1}$, Ex5$_{TRHR}$, and Ex6$_{CHRM3}$ differentially contribute to the spinal representation of allodynia. Finally, we see altered tuning of SPB neurons, which shift towards LT stimuli, thereby providing a neural substrate through which normally innocuous information could be conveyed to the brain via a nociceptive output pathway.

## Ex1$_{TACR3/1}$ neurons are the initial responders to capsaicin

The application of capsaicin to the skin activates TRPV1-expressing primary afferents, which target the most superficial aspect of the dorsal horn and release the neuropeptides SP and CGRP. Our study revealed that Ex1$_{TACR3/1}$ neurons are the primary CICADA population in the dorsal horn that responds to this afferent input. Importantly, the time course of capsaicin-induced activity in Ex1$_{TACR3/1}$ neurons matches the duration of acute pain that follows capsaicin treatment in human subjects. Compared to neurons in other excitatory populations, Ex1$_{TACR3/1}$ neurons are the most superficial, the most polymodal, and have the largest receptive field sizes (Supplementary Fig. 9). These findings suggest that Ex1$_{TACR3/1}$ neurons are likely involved in mediating capsaicin-induced pain and initiating central sensitization.

## Central sensitization involves distinct neuronal elements

We found that three CICADA-defined populations showed capsaicin-induced alterations in their response properties to low threshold mechano-sensory input and, strikingly, these neural populations changed in different ways. Neurons in the Ex6$_{CHRM3}$ population showed a significant increase in receptive field size coupled with amplified responses to most types of sensory stimuli, but no overall change in tuning. In contrast, neurons in the Ex5$_{TRHR}$ population showed a selective increase in their responsiveness to LT mechanosensory input, with no change in receptive field size. Finally, neurons in the Ex2$_{TACR1}$ population showed a combination of these two features, with both an increase in receptive field size and a shift in tuning towards LT input. These capsaicin-induced changes in activity are consistent with a model in which distinct lamina II populations mediate separate aspects of capsaicin-induced sensitization—amplification and tuning—which are integrated by interneurons in lamina I before being conveyed to the brain (Fig. 8f).

Current models of neural circuitry in the dorsal horn suggest that after injury there are a variety of different mechanisms, many of which depend on the type of injury, that could lead to an allodynia driven by central sensitization. One of the most established concepts is that allodynia is caused by disinhibition involving a loss of inhibitory neuron activity, which is especially common in the context of neuropathic injuries[2,11,12,17–24]. Other studies have suggested sensitization of excitatory pathways through mechanisms such as long-term potentiation or alterations in intrinsic excitability[26–30]. When comparing inhibitory and excitatory superficial neurons in vivo, only excitatory neurons had altered responses to low-threshold input after capsaicin[26]. This excitatory neuron specific enhancement could be driven by more ventral excitatory populations implicated in the capsaicin model of central sensitization including PKCγ cells present around the lamina II/III border, CCK cells in the deeper dorsal horn, and potentially calretinin neurons[9,14]. Because our imaging was limited to the superficial dorsal horn (lamina I–IIi), it is unlikely that we imaged many, if any, PKCγ or CCK cells. These deeper populations could be providing low-threshold input to the superficial populations we examined here through disinhibition via deep dorsal horn populations[40] and parvalbumin inhibitory neurons[22] although it was not specifically examined in this work.

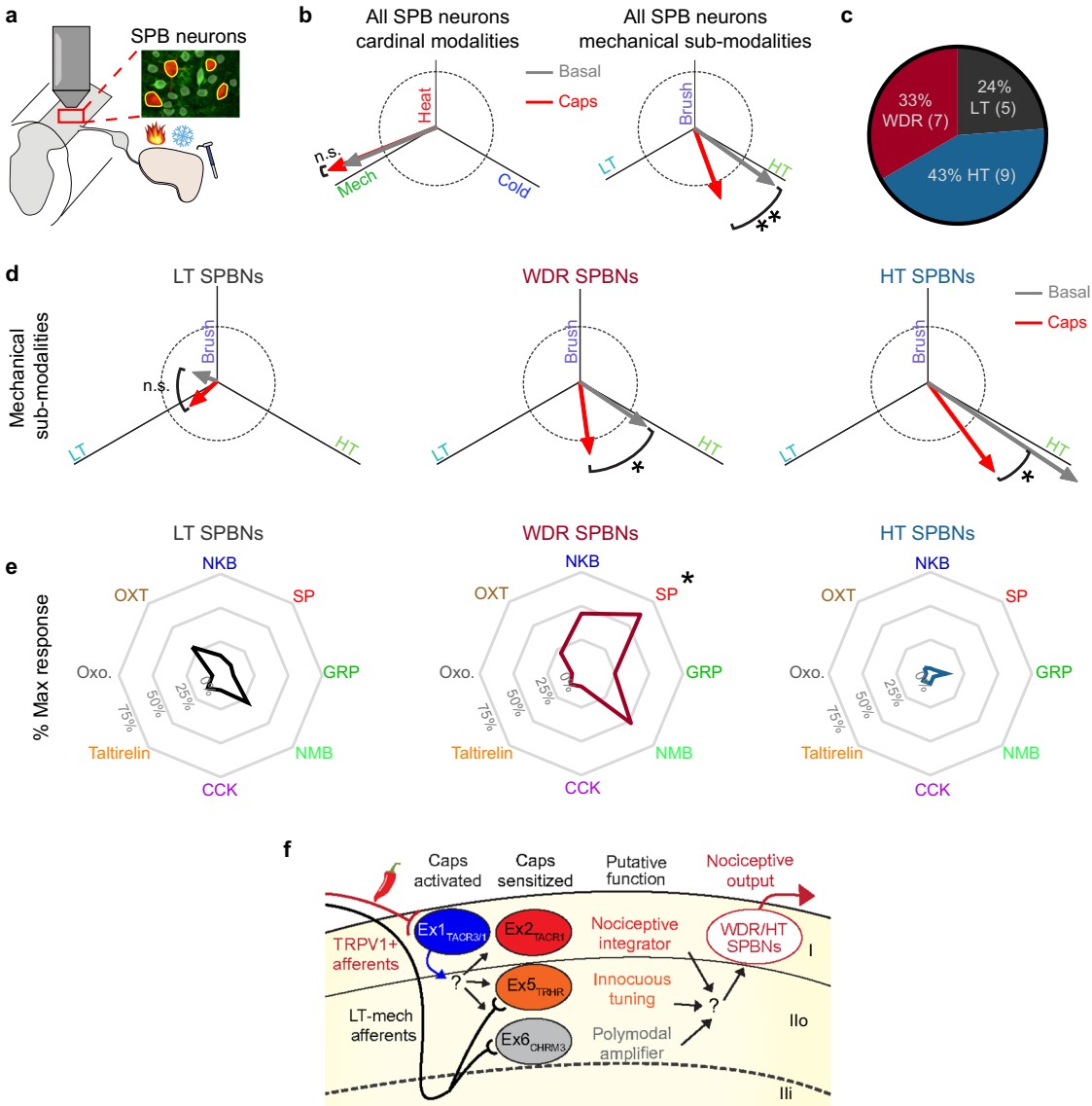

**Fig. 8 | Functional classes of SPB neurons show distinct alterations in tuning after central sensitization. a** Experimental design. **b** Left: cardinal modality (left) and mechanical sub-modality (right) tuning of SPB neurons (SPBNs) before (grey) and after capsaicin injection (red). **P < 0.01 (Two-tailed Wilcoxon matched pairs signed rank test comparing angle, P = 0.0005). **c** Pie-chart of the 3 functionally defined classes within SPB neurons. Wide-dynamic range (WDR, Red) SPBNs are defined as those that responded to both low and high intensity stimuli but encoded the stimuli intensity in their response amplitude. High-threshold (HT, Blue) SPBNs responded primarily to the 2.0 g VF filament (HT stimulus). Low-Threshold (LT, Black) SPBNs only responded to brush and/or the 0.16 g VF filament (LT stimulus). **d** Mechanical sub-modality tuning of LT (left), WDR (middle), and HT (right) SPBN sub-groups before (grey) and after capsaicin injection (red). Dotted circle is 1 unit vector in diameter. *P < 0.05 (Two-tailed Wilcoxon matched pairs signed rank test WDR: P = 0.0391, HT = 0.0117). **e** Radar chart showing the response to CICADA ligands within LT (left), WDR (middle), and HT (right) SPBN sub-groups. The spoke length is the population normalized response to CICADA ligands. There was a significant main effect of CICADA agonist response (F (3.885, 65.48) = 3.377), P = 0.015, and a significant interaction between class and CICADA agonist response (F (14, 118) = 3.143), P = 0.0003 (Mixed-effects restricted maximum likelihood model). Post hoc testing was performed comparing CICADA agonist response across functional groups, *Q < 0.05. (HT vs WDR, Q = 0.0105, LT vs WDR, Q = 0.0105). **f** Proposed model and putative roles for identified excitatory populations in the development of capsaicin induced allodynia.

## Neural correlates of psychophysical phenomena

In psychophysical studies of capsaicin-induced central sensitization, participants report two salient phenomena: touch stimuli evoke pain (allodynia), and the region of sensitivity extends beyond the initial site of injury (extended secondary zone). To help visualize the neural basis of allodynia, we developed a vector-based method to represent neuronal tuning to somatosensory stimuli. This way of visualizing the data is advantageous because it emphasizes the relative tuning of neurons to distinct types of stimuli. Because neurons in the superficial dorsal horn are rarely unimodal, we speculate that the sensory tuning may be the salient feature that is relevant for neural coding (Fig. 2). Although this method provides a distinct metric to quantify tuning, a limitation

to this approach is that the absolute response amplitude to a given stimulus is obscured.

To help visualize the neural basis that might account for the extended secondary zone caused by capsaicin induced central sensitization, we developed an approach for population-level receptive field mapping. We found that two CICADA-defined populations showed an increase in receptive field size following capsaicin treatment. Thus, the overall changes in neural activity described here may account for two of the most salient features of capsaicin-induced sensitization. Human psychophysical experiments have revealed that heat and brush responses are likewise sensitized by capsaicin, although these effects are more transient and occur over a smaller area of skin compared to

punctate mechanical allodynia[37]. A limitation of our study is that we did not capture the corresponding neural representations of thermal hyperalgesia or dynamic allodynia, likely because the Peltier thermode and brush stimulus used in this study were applied to the entire receptive field, thereby limiting our ability to see localized changes in the dorsal horn. However, using this large Peltier ensures we have covered the entirety of the skin's receptive zone, which makes the decreased response to heat after capsaicin induced sensitization within $Ex2_{TACR1}$ (Fig. 7g) as well as the reduction in amplitudes of response to heat (Supplementary Fig. 8d) particularly interesting and consistent with at least one study in non-human primates[43]. As $Ex2_{TACR1}$ is basally the most heat tuned population (Fig. 4h) and is the only CICADA population with decreases in heat responsiveness (Fig. 7), this suggests that capsaicin-induced central sensitization affects multiple modalities to shape sensory tuning rather than simply enhancing mechanical pathways.

We specifically chose an ex vivo approach as it offered several advantages over in vivo recordings because it: (1) facilitated the development of CICADA, (2) yielded hundreds of neurons per animal, which is required for population-scale analysis, (3) provided higher quality recordings due to fewer movement artifacts, (4) facilitated a more quantitative application of cutaneous stimuli, and (5) enabled recordings in the absence of anesthesia. Although this ex vivo approach has a number of appealing benefits, we acknowledge that it is a reduced preparation that lacks numerous components, notably descending modulation which has been well established to inhibit or facilitate nociceptive signaling. In addition, our ability to perform receptive field mapping is limited by the number of nerves and the size of skin attached, i.e. if a SDH neuron's receptive field would in vivo expand beyond the boundaries of the dissected piece of skin, we would not detect it and we are thus likely underestimating the degree of receptive field expansion. Nevertheless, we see that our SPB neurons have broadly similar response properties with those measured in vivo (Supplementary Table 2)[35,49,58–62]. However, it should be noted that SPB neurons generally have heterogeneous response properties, and there is significant variability across previously published studies. These inconsistencies are likely due to a variety of methodological differences such as sampling biases, anesthesia, species differences, and, to some extent, the use of qualitative stimuli such as brushing and pinching (Supplementary Table 2). If we compare our study to those that reported both force and area for their static mechanical stimuli, we find our data aligns quite well with in vivo recordings (Supplementary Fig. 4f). We observed that SPB neurons from naïve mice respond in graded fashion across mechanical intensity (Fig. 8b), consistent with prior in vivo studies[60,61]. We likewise found that, although many changes are observed following injury, the responses to innocuous stimuli show the greatest degree of change similar to prior reports[35]. Finally, we find that SPB neurons are predominantly polymodal (Fig. 2b–e), a finding consistent with existing literature, where reported polymodality in SPB neurons has ranged from 34% to 93% (average of 59%) (Supplementary Table 2)[35,49,58–61,63,64].

## Towards the modeling of somatosensory integration

Because prolonged activity in TRPV1-positive afferents gives rise to the release of SP and CGRP in the dorsal horn, it is tempting to speculate that these neuropeptides may be involved capsaicin-mediated sensitization. In this regard, it is noteworthy that many of the neurons that showed altered activity following capsaicin ($Ex1_{TACR3/1}$, $Ex2_{TACR1}$, and the WDR SPB neuron subtype) express the receptor for SP. Whether any of these populations also express receptors for CGRP remains unclear. The receptor for CGRP is Gs-coupled (rather than Gq-coupled) and, as a result, the activation of this receptor is not visible using the current CICADA strategy. The future development of tools and strategies that allow us to visualize Gs- and Gi-coupled signaling events

would not only aid in our understanding of neuromodulatory mechanisms, but also permit a much more complete analysis of neuronal subtypes through pharmacological profiling.

This study represents the first attempt to categorize neurons based on their functional responses to GPCR ligands at a population level. A key remaining question is how well can we relate our results to single-cell sequencing studies. In particular, we do not yet know to what degree the expression of mRNA will predict the functional GPCR responses we have observed here. In some cases, there appears to be a strong correspondence between our functional data and the predictions from transcriptomics. For instance, comparing our putative populations to those described in Russ et al.[13] suggests that $Ex6_{CHRM3}$ may represent Excit-1, and that $Ex5_{TRHR}$ may represent Excit-8. However, many gaps remain and future validation will be essential, particularly because mRNA expression does not perfectly predict functional responses of the corresponding proteins. Long term, the efforts to cross-reference cell subtypes that have been defined based on different experimental paradigms will be essential to create a uniform schema in the dorsal horn and enable the modeling of somatosensory integration at a new level.

## Methods

### Animals

All experiments were performed with approval of the University of Pittsburgh's Institutional Animal Care and Use Committee (IACUC, Protocol Numbers 21100045 and 21038819). Mice, from a C57/BL6 background, that were heterozygous for both the *Vglut2-ires-cre* allele (Jax Stock #016963) the *Ai96* allele (for Cre-dependent expression of GCaMP6s, Jax Stock # 028866) were used for all experiments. Mice were housed on a 12-hour light cycle with ad-libitum water and food in accordance with the United States National Institutes of Health guidelines for the care and use of laboratory animals. Relative humidity was kept between 30% and 70% and temperature was kept between 20 and 26 °C. In total 10–15 fresh air changes per hour was provided for adequate ventilation. Daily observation of all animals was required, and bedding was changed at least weekly depending on the number of animals. Social housing was utilized with no more than 4 adult mice per cage was permitted. Single housing was not used except for cases of animal welfare, e.g., aggressive males being separated. Enrichment was provided in the form of soft bedding that can be burrowed within, plastic housing domes, and running wheels. Ca2+ imaging was performed on mice of both sexes ranging from 5 to 10 weeks of age ranging from 20 to 25 grams depending on sex and age.

### Parabrachial dye injections

Mice ranging from 4 to 5-weeks old were placed under isoflurane anesthesia and mounted in a Kopf stereotaxic headframe. After exposing and leveling the skull a burr hole was made at (RC/AP: −5.2 mm from Bregma or −1.5 mm; Interaural line ML: 1.25 mm; DV: 3.0 mm from bregma or 2.23 mm from skull) and a glass micropipette filled with Fast DiI oil (2.5 mg/mL; Invitrogen, Carlsbad, CA) was used to inject 250 nl into the left parabrachial nucleus over the course of 5 min. The pipette was left in place for 10 min before retracting the pipette and closing the wound. Injected mice were then used for imaging a minimum of 4–5 days after injection. DiI-labeled SPB neurons were validated by taking fresh transverse spinal cord slices and coronal brain sections and checking for fluorescence in the appropriate regions as mapped in the Allen Mouse Brain Reference Atlas.

### Semi-intact somatosensory preparation

Mice were deeply anesthetized with a ketamine/xylazine mixture (1.75 mg ketamine and 0.25 mg xylazine per 20 g body weight). The right hind paw, leg, and back were shaved with an electric clipper

 

leaving 2–3 mm of hair in place. Next, mice were transcardially perfused with 95%O$_2$/5%CO$_2$ saturated, chilled, sucrose-based artificial cerebrospinal fluid (ACSF). The sucrose-based ACSF contained (in mM): 234 sucrose, 2.5 KCl, 0.5 CaCl$_2$, 10 MgSO$_4$, 1.25 NaH$_2$PO$_4$, 26 NaHCO$_3$, and 11 Glucose. A complete dorsal laminectomy was performed followed by excision of the spinal column (taking care not to remove tissue between the right leg and column), right ribs, and right leg. The tissue was transferred into a dissection dish and the lateral femoral cutaneous and saphenous nerves were dissected out in continuum from the skin to the DRG. The thoracolumbar spinal cord was then removed from the bone and the roots were carefully pulled away from the cord to allow movement of the spinal cord. The spinal cord was pinned at an angle such that the grey matter of the spinal cord was parallel with the ground plane. The dura mater was removed among the entire spinal cord and the pia was only removed along the L1 - L2 recording area. The roots and nerves were loosened from the tissues as much as possible and the skin was pinned out as far away from the spinal cord as possible. This ensured sufficient separation from the spinal cord as to allow a 3D printed light shield to block the visible light used to localize stimulation on the skin from reaching the high sensitivity detectors on the microscope. Once the skin and spinal cord were securely pinned in place using Minutien pins (FST Cat: 26002-20) the dish was transferred to the microscope and washed with standard normal ACSF solution containing (in mM): 117 NaCl, 3.6 KCl, 2.5 CaCl$_2$, 1.2 MgCl$_2$, 1.2 NaH$_2$PO$_4$, 25 NaHCO$_3$, 11 glucose), which was slowly washed in over 15 min. Next, the temperature of the wash solution was raised from room temperature to 28 °C in 2 °C steps every 5–10 min. Once the tissue had equilibrated to the standard ACSF, a manual wash was performed to fully remove any leftover detritus and dissection solution using a 50 ml syringe filled with ACSF. Imaging began 30–45 min after the final wash in order to allow time for thermal expansion and settling of the tissue. Once ~1 L of solution was washed through the dish, the remaining 4 L of bubbled ACSF was then recirculated for the duration of the physiology experiments.

## Multiphoton imaging

Ca$^{2+}$ imaging was performed on a Leica SP5 MP microscope using a Leica 20x NA 1.0 water immersion lens with a Coherent Chameleon Ultra II Ti:Si femtosecond laser set to 940 nm for Ca$^{2+}$ imaging and 1040 nm for imaging of DiI within the SPBPNs. Fluorescence was captured with HyD detectors and standard FITC/TRITC emission filter sets with a 570 nm dichroic beam splitter. Three different planes of the superficial grey matter were simultaneously imaged with a spacing of 14 μm between planes, which yielded a range of depths from 0 to 60 μm below the spinal cord surface. For each plane, scanning was performed at 1 Hz with a 1.4x optical zoom at 512 × 214 pixel resolution and a field size of 528.2 × 220.8 μm size yielding a pixel resolution of 1.03 μm/pixel. This volumetric scanning allowed sampling from multiple lamina simultaneous and typically yielded ~300 excitatory neurons in a given experiment.

As the location of afferent input for each animal is determined by relative somatotopy, the region of the grey matter imaged thus needed to be functionally identified for each animal. The exact field of view imaged in the SDH was determined by using a low magnification image and a search stimulus within the dissected skin. A wide brush which stimulated the entire length of the skin was used to activate as many afferent inputs as possible. The imaging field of view was iteratively refined, centering the image on the region that had the most change in fluorescence and then zooming to the final 1.4x optical zoom. Subsequently, a 2.0 g von Frey (VF) filament was used to delineate a 15 × 15 mm region of skin that showed the most input into the final field of view in the SDH. The region was marked with a surgical felt tip pen. Imaging, stimuli controllers, feedback sensors (thermocouples, load cells, etc.), and event times were synchronized using a Power1401 and Signal (CED).

## Drug applications

After the final round of sensory testing post capsaicin was completed, 500 nM tetrodotoxin (TTX) was applied and allowed to circulate for 15–20 min before proceeding. CICADA agonists were then applied in a randomized order for 2 min followed by 5 min wash in the continued presence of 500 nM TTX. CICADA agonists were kept frozen in single use aliquots until application when they were thawed and diluted in TTX recording solution. Post CICADA, a brief pulse of 5 ml of ACSF containing 30 mM KCl ACSF solution was injected into the perfusion lines in order to confirm cell viability and assist in ROI generation and refinement. All drugs were purchased from Tocris and Sigma.

## Determination of dorso-ventral location

In order to accurately calculate the depth of cells in the dorsal horn, the curved surface of the dorsal horn needed to be taken into account. Briefly, after all experimental manipulations had finished, a volumetric time-lapse (XYZT) imaging protocol was loaded at the same field of view and XY resolution as the Ca$^{2+}$ imaging protocol but encompassing the entire recorded volume of the grey matter at high Z resolution taken from -10 μm above the surface to -60 μm below, at a Z resolution of 1 μm. During this time, 30 mM KCl ACSF solution was applied in order to visualize all the GCaMP6s expressing cells and provide a high contrast image for volumetric reconstruction. Post-hoc, the structural reference was aligned with the functional Ca$^{2+}$ imaging reference image, smoothed with a 3D median filter, contrast enhanced using Contrast Limited Adaptive Histogram Equalization, and then XY down sampled into voxels representing a 10 × 10 × 1 XYZ area. The mean value of the voxels was then used to find the surface of the dorsal horn grey matter for each 10 ×10 XY coordinate. The calculated Z coordinate of the surface was based on identifying the asymptotic increase in fluorescence as the surface of the dorsal horn comes into focus. Each of the 3 recording planes was then aligned to a Z-plane in the structural image and each ROI was then offset in the Z-dimension by the amount indicated by their XY location. This analysis provided the final dorso-ventral location of each cell.

## Natural cutaneous stimuli application

The presentation order of natural stimuli was randomized within each testing block (i.e. baseline or post capsaicin testing periods). All natural stimuli were applied to an identical 15×15 mm area of skin using stimulators that were specifically designed to fit in the same footprint. A dim red light was focused on the skin in order to provide sufficient illumination for the experimenter and reduce the amount of light contamination reaching the sensor. For manually applied stimuli, a program was written in Signal (CED) to produce a synchronized output to an audio monitor that provided a 3,2,1 countdown for each stimulus, a 1 s tone for duration of application, and ensured both even spacing and duration of stimuli application.

## Mechanical stimuli

Airpuff was applied at a constant supply pressure of 60 PSI for 1 s duration and repeated 3 times 60 s apart and controlled by a PicoSpritzer III (Parker Hannifin). The airpuff applicator device was a custom designed module with a 6 ×6 grid of evenly spaced 18-gauge stainless steel nozzles with an inner diameter of 0.8 mm to fit within the delineated region (15 × 15 mm) and provide equal pressure across the skin. The device held in place 1 cm above the skin with a micromanipulator. Brushing was done manually with a quantitative sensory testing brush (Somedic SenseLab AB, SENSELab Brush-05). Average force applied was ~20 g for brushing and ~5 g for airpuff and was calibrated by use of a load sensor coupled to the imaging chamber and controlled by an Arduino. As brushing was manual, we did observe some variance in the force applied but it typically stayed within the range of 15 to 25 g. The applied pressure for each mechanical stimulus was estimated as follows: air puff, 0.2 kPa; brush, 6.5 kPa; 0.16 g VF (LT), 78 kPa; 2.0 g VF

(HT), 270 kPa. While the Somedic brush is a 15 × 5 mm brush with an area of 75 mm², we estimated the thickness of the brush when compressed during a stroke was ~2 mm resulting in an effective area closer to 30 mm² when actively applying pressure. The brushing was repeated 3 times, 60 s apart.

Receptive field mapping was done with a LT (0.16 g) or HT (2.0 g) von Frey filament. A custom cut flexible plastic grid was placed upon the delineated region of skin in order to assist with consistent targeting of the filaments to each of the 16 sites to be mapped in a 4 ×4 grid arrangement. Each round of mapping consisted of applying the filament for 1 s to each location spaced 15 s apart. For example, in Fig. 3d, the first region stimulated was A1 then B1, C1, D1, A2, B2, etc. Once the final location was reached the mapping was repeated once more for a total of 2 replicates for each filament and testing block.

## Thermal stimulus application

A custom-made water-cooled Peltier was placed on the delineated region of skin and held at a thermoneutral 30 °C between stimulations. The Peltier was then driven by a custom-designed Peltier controller to the indicated temperatures at specified rates and set point. Measured surface temperatures were recorded in Signal 7 (CED) with a CED 1401. For fast rate heat ramps the Peltier was driven at 4 °C/s from 30 °C to 52 °C and held for 5 s at 52 °C before returning to neutral. Fast cold ramps were similarly run at 4 °C/s but the effective cooling rate slows as it approaches 0 °C (as seen in Fig. 1i). Thermal steps were held for 15 s at the intermediate temperature (40 °C for heating and 18 °C for cooling) and stepped to the subsequent temperature (52 °C or 4 °C) at maximum ramp rate and held for 5 s once at the final temp. Both heating and cooling slow ramps run at a constant rate of 1 °C/s from either 30 °C to 52 °C or 30 °C to 4 °C. All thermal stimuli were applied twice for each testing block separated by at least 1 min.

## Capsaicin injection

After the first testing block was completed, 7.5 μg of capsaicin dissolved in 10 μl of PBS containing 0.5% Tween-80 was intradermally injected with a 31-gauge needle to the result of a visible bleb. Because additional light was needed for consistent needle placement, imaging was stopped during this time to prevent PMT damage, resulting in ~30 s blind spot around the injection time. The next block of sensory testing was performed 5 min following injection, followed by repeated blocks of testing ~30 min apart. The effects of capsaicin on mechanical activity were seen at least 15 min after capsaicin injection and remained consistent for the 3 rounds of post capsaicin testing performed. For analysis here, the middle block of capsaicin testing (~30 min post injection) was used for all animals.

## Image processing and data extraction

For image processing, we utilized a Suite2p (v0.10.0, HHMI Janelia) pipeline empirically tuned for superficial dorsal horn cells for image registration and initial segmentation. The masks were transferred to FIJI (V1.53i, ImageJ2) and additional ROIs were added by hand as the heterogeneity in morphology and size in the superficial layers of the spinal cord was difficult for most segmentation programs to accurately capture. Masks were then spot checked using a maximum activity (ΔF/F or the first derivative of fluorescence across time) reference image generated from 5-min bins along the entire recording. This maximum activity reference image resulted in ~50 frames that represented 5-min bins of activity and could be visually scrolled through to quickly check the 4- to 5-h long recording for physical deformations, drift, blebbing, or other pathologic changes. Any cells that showed significant deviation from their XY location, signs of Z-drift (e.g., changes in the appearance of the nucleus, soma, or processes), blebbing, or cell death in the latter portions of the recording were manually deleted. Once the masks were finalized, the mean fluorescence was calculated and any cell that has significant drift (>200%) in their 5-min binned median

fluorescence value was deleted because such a change was indicative of either physical drift or Ca²⁺ dysregulation which would signal signs of apoptosis. Then ΔF/F was calculated in Excel (Microsoft) for each cell using a rolling ball baseline: for each frame ($F_i$) the 30th percentile value of the surrounding 5 minutes of frames was used as a baseline ($F_b$) to perform the ΔF/F calculation (($F_i$- $F_b$)/$F_b$). Rolling ball normalization was preferred here as the effects of slow photobleaching of the indicator and/or small changes in location were minimized in these long recordings.

## Analysis of Ca²⁺ activity

Once ΔF/F values had been generated, traces were analyzed for binary responder analysis (yes/no) as determined by a peak value in ΔF/F that was both >125% of baseline in the 5 s following stimulation and also > 6σ (standard deviations) of basal trace noise in the 30 s prior. These criteria were selected based on a receiver operator characteristic (ROC) analysis of σ and amplitude using a manually annotated data set of 500 cells as the 'ground truth' against which we optimized the AUC from the ROC analysis. For the manual annotation, we generated image files for each cell with stimulation event markers. The 500 traces were then manually marked as a responder or not based on whether the amplitude of response was clearly distinguishable over the background recording noise. This 'ground truth' data was then used to calibrate against multiple objective measures of quantification (Supplementary Fig. 3). We chose to slightly favor specificity over sensitivity for these analyses to minimize false positives. The responses to CICADA agonists used an identical amplitude and σ criteria but with a 120 s window of response to match the duration of application. This window also accommodates the higher variation in onset of response due to the pharmacokinetics of drug penetration and variations in intracellular signaling cascades of the different agonists. Radar charts showing the % maximum response for CICADA ligands, as in Figs. 4e, 8e are min to max normalized responses of responsive cells. For cells which responded to at least one CICADA ligand (all cells in Fig. 4e are CICADA responsive by definition vs Fig. 8e includes non-responders), each cell's responses had the minimum response to any CICADA ligand subtracted to make the smallest value 0 and then the data was divided by the maximum response of that cell, *i.e.* taking the 8 data points for amplitudes of response to CICADA, the formula is as follows: ($Cell_{Response}$ - $Cell_{min}$)/$Cell_{max}$. For cells in Fig. 8e which did not respond to any CICADA ligands, normalizing by a very low maximum did not place the data in the correct context so, if a cell did not respond to any CICADA ligand it was instead of being normalized to the cell's maximum response it was instead divided by the average response to all CICADA ligands so that we place sub-threshold responses within the appropriate context and not emphasizing noise among non-responders. Data were then averaged within each indicated population to generate the % maximum response. Euler diagrams were made using eulerAPE[65].

Once the activity had been binarized, responses to the various stimuli were assigned. For brush and airpuff stimulation, a cell was considered a responder if it met threshold criteria for at least 2 of 3 applications to be considered a responder. For heating, cooling, or static LT/HT stimuli if a cell responded to at least 1 of the stimuli it was considered a response. This was due to the fact that each of the heating and cooling ramps have distinct parameters (*e.g.*, fast vs slow rate of change) and the static LT and HT stimulation each stimulate a distinct region of skin so each of the stimulations were not equal. To calculate the size of the receptive field, the total sum of responses at the 16 tested fields (4 × 4 grid) were averaged across 2 trials providing a possible receptive field size of 0 to 16. As each of the 16 tested sites were spread across a 225 mm² region spaced apart by ~3.75 mm, each response likely represents ~3-4 mm² of skin, *i.e.* a cell which responded to 4 points has an estimated receptive field size of 15 mm². For clarity, the conversion to square mm² was not performed and was left as number of receptive fields. Of note, while changes in LT receptive field

size post capsaicin are largest in Lamina II cells (Supplementary Fig. 8f–h) the per animal variability in the change post capsaicin was not well explained by a difference in average sampling depth and may represent another unknown source of variability. For capsaicin injection, a cell was considered responsive to capsaicin if the activity post capsaicin injection was greater than 150% of pre-capsaicin levels based on a 2-min average of ΔF/F values.

## Vector analysis

Vectors were calculated using the maximum amplitude of response across all trials for the indicated modality. For each cell, the 3 modalities to be compared by vector analysis were first normalized to the average response. The angle of the vector was assigned 120° apart from each other modality as follows: heat/brush (+90), mechanical/LT Static (+210) and cold/HT static (+330). These angles (e.g. +90) are relative to the positive X-axis moving in a counter-clockwise direction. Using these angles and the normalized amplitudes, 3 separate vectors were calculated, summed, and then graphed in Matlab (R2021a, Mathworks) where the length of the vector was the normalized amplitude of response, and the direction of each vector was as described for each modality. For examples in interpreting these vectors we can examine Cells #1-3 from (Fig. 2g, more examples in Supplementary Fig. 10). Cell #1 has a very clear and strong tuning for mechanical stimuli. The magnitude and angle of the bi-modal mechanical and heat responsive Cell #2 indicates a tuning for heat over mechanical stimuli. In contrast the trimodal Cell #3 shows no tuning. Cells which did not have any responses prior to capsaicin were not included in the vector analysis. For statistical tests which compare angles, it is necessary to standardize the datasets before testing such that the angles are all calculated relative to the same angle. For example, if a cell is at +1 and another is at −359 they are essentially the same angle (despite the large nominal difference this is circular data) and should be standardized such that they are all being calculated as 'degrees from axis' (i.e. they should be calculated instead as +1 and −1) such that they represent a more normally distributed data.

## K-Means clustering

K-Means clustering was performed in the statistical package Orange[57]. Normalized amplitudes of CICADA responses were used as an input for the K-means calculation and the data was visualized using t-Distributed Stochastic Neighbor Embedding (t-SNE)[66].

## Regression analysis of polymodality

To determine whether dorso-ventral location (depth relative to the dorsal surface) or CICADA population best predict cardinal polymodality, we performed a multiple logistic regression in GraphPad Prism 9 software where the outcome was cardinal polymodality and we included depth and CICADA population as variables. For Supplementary Fig. 4h, i, preference score is the average modality amplitude subtracted from an individual modality amplitude and then the product is divided by the maximum amplitude of response: $(X_i - X_{avg})/X_{max}$.

## Fluorescence in situ hybridization, imaging, and quantification

*Vglut2 cre* x *GCaMP6s* animals were anesthetized with isoflurane and decapitated. Within 5 min of euthanasia, the L3/L4 spinal segments were removed and placed in OCT, flash frozen, and kept on dry ice until cryosectioning. Spinal cord was then sectioned in the transverse plane at 15 μm thickness, mounted on slides, and were probed for *GCaMP6s* and *Vglut2* using RNAscope Multiplex Fluorescent Assay (Advanced Cell Diagnostics, 320851) using: Mm-Slc17a6 (Cat No. 319171) for *Vglut2* and Mm-GCaMP6s-O1 (Cat No. 557091) for *GCaMP6s*. After hybridization and amplification slides were mounted with Prolong Gold with DAPI. Sections were then imaged using an upright epifluorescence microscope (Leica THUNDER Imager Tissue) using a 20x/0.80 NA air lens at 3883×3886 resolution (0.32 μm per pixel) and analyzed offline with FIJI. A positive cell required a clearly defined nucleus and fluorescent signal forming a ring around the nucleus, as shown before[67]. 60-76 GCaMP6s+ nuclei were quantified within the superficial dorsal horn per mouse.

## Statistical analysis

Data are expressed as the mean ± SEM unless otherwise indicated in figure legend. The following tests were used for statistical analyses: Student *t* test (paired where appropriate *e.g.* before and after capsaicin) for comparison of exactly 2 groups, ordinary 1-way analysis of variance (ANOVA) with multiple comparisons post hoc test if indicated by the main effect (comparison of >2 groups), 2-way repeated measures ANOVA (restricted maximum likelihood (REML) was used where balanced repeated measures data was unavailable) with multiple comparisons post hoc test if indicated by the main effect (time course with comparison of 2 or more groups). All tests were 2 tailed and a value of $P < 0.05$ was considered statistically significant in all cases. For cases where multiple post hoc comparisons were indicated by the main effect, a correction for multiple comparisons was implemented using a False Discovery Rate of 0.05 via the two-stage step-up method of Benjamini, Krieger and Yekutieli. Reported *Q*-values are the FDR corrected *P*-values, and in all cases $Q < 0.05$ was considered significant. For all data plotted as a function of depth (*e.g.* Fig. 3h) smoothing was applied using Prism (9 neighbor, 6th order polynomial). We estimated our sample size with G*Power after 2 experiments were completed. All other statistical analyses were performed using GraphPad Prism 9 software.

## Reporting summary

Further information on research design is available in the Nature Research Reporting Summary linked to this article.

## Data availability

All data analyzed for this study are included in the article, supplementary figures, tables, and source data. The raw image files for all experiments are available at the following GIN (G-Node Infrastructure) repository: https://gin.g-node.org/warwickc/CICADAv1 Caps. Parabrachial nucleus injections were verified by utilizing the annotated mouse anatomical database in the publicly available Allen Mouse Brain Reference Atlas (https://mouse.brain-map.org/static/atlas). Comparisons to in Russ et al.[13] were made using the publicly available harmonized atlas of mouse spinal cord cell types (https://seqseek.ninds.nih.gov/). Source data are provided with this paper.

## Code availability

All code generated by this work for image processing and analysis are located at: https://github.com/cawarwick.

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

## Acknowledgements

The research reported in this publication was supported by the National Institute of Neurological Disorder and Stroke of the National Institutes of Health under Award Number R01 NS096705 to H.R. Koerber, NS110155 to T.D. Sheahan, NS073548 to C.A. Warwick, and the National Institute of Arthritis and Musculoskeletal and Skin Diseases of the National Institutes of Health under Award Number R01AR063772 to S.E. Ross.

## Author contributions

C.A.W., H.R.K., and S.E.R. conceptualized the project. C.A.W., J.S., T.D.S., and H.C. provided data curation. C.A.W., J.S., T.D.S., H.C., and J.I., performed formal analysis. H.R.K and S.E.R. acquired funding. C.A.W., J.S., J.H., K.M.S., and T.D.S. were involved in the collection of data for the investigation. C.A.W., J.S., T.D.S., J.I., H.R.K., and S.E.R. designed the methodology. C.A.W., H.R.K., and S.E.R. administered the project. C.A.W., J.S., J.H., K.M.D., T.D.S., and J.I. provided resources necessary for the investigation. C.A.W. and J.S. implemented code to handle and organize image files. H.R.K. and S.E.R. were the primary supervisors. C.A.W., J.S., H.R.K., and S.E.R. were involved in the validation of the experiments and data analysis. C.A.W., K.M.S., and T.D.S. created the figures and data presentation. C.A.W. wrote the original draft. All authors were involved in reviewing, editing, and the approval of the final manuscript.

## Competing interests

The authors have no conflicts of interest to declare.
