## [Peer Review File · Nature Communications]

Cell type-specific calcium imaging of central sensitization in mouse dorsal hornREVIEWER COMMENTS

Reviewer #1 (Remarks to the Author):

Using an ex vivo prep Warwick et al. have performed an extensive chemogenic profile of spinal neurons in the dorsal horn that appears to mediate capsaicin-induced central sensitization. This work builds upon previous studies that have used this group's ex vivo skin/spinal cord preparation (Hachisuka et al. 2016 and Choi et al. 2020). The studies presented here by Warwick and colleagues are well put together and detailed, with extensive findings and appropriate controls. This work provides further evidence for how certain populations of dorsal horn neurons contribute to central sensitization.

My only major concern is the lack of any testing of the functional consequence of their findings. The authors noted that the largest capsaicin-induced change was observed in neurons that responded to oxotremorine M, an agonist for the muscarinic acetylcholine receptor M3. Would inhibition of this population prior to capsaicin injection lead to changes in the other populations identified? Would the application of TACR1 antagonists affect the observed changes the authors see in the other defined populations? For example, a question might be something on the order of is Ex1TACR3/1 necessary and sufficient to modulate capsaicin central sensitization, or is one of the other populations identified necessary? Perhaps one population is responsible for thermal but not mechanical alterations? These experiments do not need to be extensive and would greatly strengthen the author's conclusions.

One other minor point. The authors used von Frey filaments of 2.0 g for high threshold and 0.16 g for low threshold stimulation. For mice, at least with regards to hindpaw stimulation, high and low thresholds stimulation is generally much lower, 0.07 g and 0.4 g respectively. The skin prep the authors use incorporates the dorsal hind paw and proximal hip regions. Do these areas require greater von Frey stimulation to evoke a response or is it merely a consequence of the prep and the loss of underlying muscle and fascia?

Reviewer #2 (Remarks to the Author):

This paper mapped somatosensory receptive fields and modalities of spinoparabrachial (SPB) neurons and excitatory spinal interneurons in dorsal horn laminae I and 2 on a populational level by imaging Ca²⁺ activity using GCaMP6s and multiphoton microscopy both before and after capsaicin application to visualize central sensitization. This is a step towards understanding central sensitization with a long-term goal of understanding how, and what cellular mechanisms central sensitization turns acute pain to chronic pain. Neurons are divided into classes by mapping specific Gq protein-coupled receptors in a process the authors named CICADA (Cell-type Identification by Ca²⁺-coupled Activity through Drug Activation). Neurons were categorized by lamina position, CICADA, sensory modality, and response to capsaicin application.

Strengths: Population level analysis of spinal cord somatosensory neurons is lacking to date and this paper represents a significant step forward in this area. Authors perform paired analysis of neurons before and after capsaicin application. Anesthesia confounds of these kinds of experiments are avoided. Neurons are classified by GPCR expression pattern, allowing researchers building on these results for clear cellular markers. Extended data shows a great deal of testing to determine limits and reproducibility of these methods.

Limitations: Authors acknowledge many limitations. Most somatosensory neurons were not labeled by CICADA which is limited to GPCRs that stimulate Ca²⁺ release from intracellular stores. Authors point out their method can be adapted to other forms of labeling that can detect different kinds of GPCRs. Imaging and analysis are labor and time intensive, limiting animal number. Ca²⁺ imaging limits analysis to superficial lamina of the dorsal horn, but extended data suggests deeper imaging may be possible. It might be better authors develop this imaging method in vivo.

Major Concerns

Given the most polymodal CICADA populations (Fig.4h) were also the most skewed towards lamina I (Fig.4f), I would like the authors to discuss more of the degree to which tuning can be predicted

by CICADA category v. anatomical position. I.e., could the polymodality of Ex1, Ex2, and Ex3 be better predicted by almost entirely confined to lamina I? How does the tuning of unlabeled lamina I neurons compare to Ex1, Ex2, and Ex3?

Authors analyzed lots of vector direction but should not explain why they do vector analysis and what each vector analysis means for results, central sensitization, or chronic pain, and sensitization. Would you please extensively articulate this for each vector analysis and figure? Many experiments use very low or minimum animal number. Should increase at least $n = 9$ or higher.

Two animals showed large changes in low threshold receptive field sizes following capsaicin and two showed modest increases (Fig.6f). Are these differences a result of different numbers of neurons labeling in different lamina? I.e., did large changes correspond to animals having a higher percentage of labeled neurons in lamina I? A sentence or two in discussion and/or results could clarify this issue.

Minor concerns

P18, line 413, second paragraph of Multiphoton Imaging section "filament was used to delineate a 15 x 15 mm region of skin was delineated that showed" Delete the crossed out words.

Figure2 label is wrong, Figure2i should be h.

Reviewer #3 (Remarks to the Author):

In this manuscript, Warwick and collaborators investigate the spinal dorsal horn network with calcium imaging to identify populations mediating capsaicin-induced central sensitization. To do so, they combine an ex vivo preparation that enables recording calcium activity at the cellular level with "natural" peripheral stimulation, with a pharmacological profiling strategy that enables a novel classification of dorsal horn neurons. Overall, the manuscript is very clear and well written, the experiments and their controls are adequately presented. However, it provides only limited new functional insights and omits discussing the abundant literature on dorsal horn networks, and on unmasking of low threshold inputs to neurons that normally respond to high threshold inputs (both in vitro and in vivo).

Main concerns:

* The authors claim (l. 53) that only few studies have applied Ca^{2+} imaging to the spinal cord. They certainly refer to "in vivo" experiments, as there are countless in vitro calcium imaging studies. As for in vivo, citing only two of them is also misleading. From Johannssen & Helmchen's paper in JPhys 2010 (doi: 10.1113/jphysiol.2010.191833) to the in press Sullivan & Sdrulla one (JNeurosc., 2022, DOI: <https://doi.org/10.1523/JNEUROSCI.1860-21.2021>) many papers used this approach in vivo in anesthetized mice (e.g. Nishida et al. PLOS one, 2014, <https://doi.org/10.1371/journal.pone.0103321>). Interestingly, performing spinal cord calcium imaging in freely behaving mice has proven possible by the Nimmerjahn's lab in 2016 (NatCommun. 2016, DOI: 10.1038/ncomms11450). In contrast, the ex vivo preparation used here has limitation that should be discussed.

* The authors have leveraged the spinal cord ex vivo with attached limb preparation for electrophysiological recordings in a previous publication (eLife 2016, <https://elifesciences.org/articles/22866>). Here, they use it for calcium imaging with the addition of a profiling approach called CICADA. The profiling approach is certainly interesting, as other dorsal horn calcium imaging studies only distinguish excitatory/inhibitory (Sullivan & Sdrulla, JNeurosc., 2022) or projection neurons (Chisholm et al. Pain 2021, doi: 10.1097/j.pain.0000000000002226). However only 25% of excitatory neurons are CICADA responders, which implies that some conclusions should be tuned down (see specific points below).

* Spinoparabrachial neurons have been previously studied with in vivo calcium imaging (Chisholm et al. Pain 2021) or in vivo electrophysiology (e.g. Keller et al., MolPain 2007, doi: 10.1186/1744-8069-3-27). There is no comparison of the present results with these older studies, while for example the % of different cardinal modalities or polymodal neurons differs and should be

discussed.

* More generally, dorsal horn neurons have long been classified, either based on their electrophysiological profile, their morphology or more lately based on genetic markers. A significant contribution to the field would imply to validate the new profiling proposed here with at least one of the textbook classifications (see also next point).

* This is even more true as the authors focus on capsaicin-induced plasticity, a subject that has been abundantly studied in vitro. Based on patch-clamp studies of capsaicin-responding neurons (directly or indirectly), several labs have proposed dorsal horn circuits models (among many others: Petitjean et al., EJM 2012, <https://doi.org/10.1111/j.1460-9568.2012.08273.x> ; Lu Y et al. J Clin Invest 2013, DOI: 10.1172/JCI70026 ; Ganley et al., JNeurosci 2015, DOI: 10.1523/JNEUROSCI.0406-15.2015). More recently, it was shown that the activation of calretinin neurons that are capsaicin responding and presynaptic to parabrachial neurons, induces allodynia and spontaneous pain (Petitjean et al., CellReports 2019, doi: 10.1016/j.celrep.2019.07.048. PMID: 31390558). Again, this literature should be cited and discussed, and attempts should be made to link the present results with what is already known on this circuit.

* The unmasking of low threshold inputs to neurons that normally respond only to high threshold inputs, as a substrate for allodynia, has long been discussed and demonstrated in the field. While some labs have focused on the disinhibition enabling this plasticity both in vitro or in vivo (Torsney & MacDermott, JNeurosci 2006, doi: 10.1523/JNEUROSCI.4584-05.2006 ; Lavertu et al., Brain 2014, <https://doi.org/10.1093/brain/awt334> ; Medrano et al., Pain 2016, DOI: 10.1097/j.pain.0000000000000538), others have investigated the underlying dorsal horn network: we mentioned above the implication of calretinin neurons (Petitjean et al., CellReports 2019) but NPY, PKCgamma or CCK neurons have also been implicated in neuropathic models (Tashima et al. PNAS 2021, DOI: 10.1073/pnas.2021220118 ; Peirs et al., Neuron 2021, DOI: 10.1016/j.neuron.2020.10.027). Importantly, the response profile of spinoparabrachial neurons has been shown to shift towards low threshold responses in a neuropathic pain model (Keller et al., MolPain 2007), while capsaicin was shown to markedly enhance the responses of excitatory neurons to innocuous touch (many neurons gaining sensitivity) (Sullivan & Sdrulla JNeurosc., 2022).

* A last general comment is that presenting only 20 references is difficult to justify considering all this valuable literature.

Specific and/or minor points:

l. 48: have revealed "that" there are

l. 64: heterogeneous expression "of" Gq-coupled G-proteins

l. 89: document (here or in the Methods) the specificity of Cre expression in these mice

l. 128-130: The vector-based method is interesting to reduce the dimensionality of the data. However, its limitation should be discussed. In particular, a vector pointing to the HT direction could either reflect a specific HT neuron or a polymodal neuron with similar amplitude LT and heat response. As there are only 3 initial dimensions, one may wonder if a classical 3D representation would not be sufficient to visualize and analyze the data without loss of information.

l. 145: include references for the somatotopy of DH organization

l. 149: include here a brief description of receptive field size quantification approach

l. 171 and following: indicate that the order of agonists application has been controlled for

l. 179: As mentioned in the Main concerns, the low % of CICADA responding neurons is an issue. Extended data Fig. 5f shows that this is highly variable depending on the depth; this result should be in the main text and discussed.

l. 185: cell clusters showed "qualitatively different" distributions according to depth; there is no statistical difference presented

l. 191: Authors propose a classification of DH neurons according to their cardinal modalities (Fig. 2g); did they attempt to correlate this classification to the one emerging from CICADA (Ex1-7)?

l. 211: Ex1 was the only "CICADA" population that showed statistically significant predictive power for capsaicin response; the vast majority of excitatory neurons are not responding to CICADA.

l. 215-216: again, among CICADA responding neurons, Ex1 is predominately driven, but other non-CICADA responding neurons may be strongly driven, this is not discussed.

l. 220: These long experiments raise two questions:

* what is the stability of cutaneous stimulation responses without capsaicin treatment?

* CICADA is performed at the end: could possibly the application of capsaicin induce a plasticity in the CICADA profile?

l. 222: remove "to LT stimulate" (said in l. 221)

l. 251-260: A striking observation for the plasticity of Ex2 is also the fact that they are less responsive to Heat after capsaicin according to Fig. 7g. This should be presented in the results and discussed.

l. 548: Why is the definition of capsaicine responsive neurons so different from CICADA?

Fig. 2: where is 2g? In 2i, the "brush static LT" population is mislabeled N=1265 cells but how many SPB?

Fig. 3: Indicate number of neurons in 3e, indicate briefly how RF size is quantified
How is 3h built? It does not look like the RF size/depth of all of the neurons that would look more like a points cloud. Is it an average RF size for a certain depth interval? The variation should be presented if this is the case.

3i: again, if each point represents a mouse, authors should illustrate mean and dispersion.

Fig. 4: Is the color code in 4b the one used for the rest of the figure? Is it the same cell represented with a given color in all of 4b panels? The blue cell per example (Ex1??) has a strong response to Taltirelin which does not fit with 4d.

In 4h, why use two tests to compare the vector angle and magnitude? A two-way ANOVA could be used to take into account these two aspects of the vector.

Fig. 5: A scale is missing in 5f for RC/LM distance

Fig. 6: N=4 mice in 6e?

In 6f one expects an average and dispersion for the RF size by animal.

Fig. 8: Why use the "old" terminology in 8c while they have presented another new and quantitative way of classifying neurons according to their cardinal modality in 2f?

How is the distribution in 8c modified after capsaicin?

What is the "max response" used to normalize the data in 8e?

Extended Data Fig. 4a: Top left panel presents red only, so the green labeled is not needed. Middle left panel: what stands the left panel "ventral" for?

Legend of panel g,h: for the preference score, why divide by Xmax and not Xavg?

For the presentation of tuning angle vs. depth (panels I and k), as in 3h, one expects either a points cloud, or an average per depth interval (detailed in the figure legend) with the deviation.

Panel i: only the most superficial cells showing "some" tuning for heat; their response is most tuned for mechanical stimulations.

Legend of panel j ("In particular, the most superficial cells are more tuned for heat, Lam IIO cells

tend to have high 840 amplitude responses (high magnitude vector), whereas Lam III begins to show smaller 841 magnitude mechanical responses relative to other modalities all while following the general 842 trend of preferring mechanical stimuli at deeper lamina") looks like a strong assertion but lacks statistical analysis.

Blue text indicates the authors response to reviewers

Red text indicates additions or major changes in the text of manuscript

Reviewer #1 (Remarks to the Author):

Using an ex vivo prep Warwick et al. have performed an extensive chemogenic profile of spinal neurons in the dorsal horn that appears to mediate capsaicin-induced central sensitization. This work builds upon previous studies that have used this group's ex vivo skin/spinal cord preparation (Hachisuka et al. 2016 and Choi et al. 2020). The studies presented here by Warwick and colleagues are well put together and detailed, with extensive findings and appropriate controls. This work provides further evidence for how certain populations of dorsal horn neurons contribute to central sensitization.

1. My only major concern is the lack of any testing of the functional consequence of their findings. The authors noted that the largest capsaicin-induced change was observed in neurons that responded to oxotremorine M, an agonist for the muscarinic acetylcholine receptor M3. Would inhibition of his population prior to capsaicin injection lead to changes in the other populations identified? Would the application of TACR1 antagonists affect the observed changes the authors see in the other defined populations? For example, a question might be something on the order of is $Ex1_{TACR3/1}$ necessary and sufficient to modulate capsaicin central sensitization, or is one of the other populations identified necessary? Perhaps one population is responsible for thermal but not mechanical alterations? These experiments do not need to be extensive and would greatly strengthen the author's conclusions.

Although we agree with Reviewer 1 that it would be nice to test the functional consequence of our findings, we submit that the tools to do such an experiment are not currently available. To our knowledge, there are no genetic tools that would selectively target any of the identified populations. Furthermore, pharmacological tools would be unlikely to provide insight for two reasons: (1) the antagonists *would not be specific* to the identified cell types, and (2) the antagonists *would not necessarily inhibit* the targeted cells, even if they were specific. As a result, the resulting data would be difficult to interpret.

Lack of specificity: TACR1 antagonists would not only target $Ex1_{TACR3/1}$ but would also affect $Ex2_{TACR1}$ and spinal projection neurons that express TACR1. Similarly, the muscarinic acetylcholine receptor is broadly expressed in inhibitory neurons as well as $Ex6_{CHRM3}$ neurons. In each case pharmacological antagonists would target multiple populations which would prevent us from assessing whether a specific population is necessary for central sensitization.

Lack of inhibition: Even if the receptors were specific to a specific cell type, the proposed experiment would still be hard to interpret because we would have no way of knowing to what degree the receptor antagonists are inhibiting the cells of interest. For instance, muscarinic receptor antagonists would be expected to inhibit acetylcholine signaling. Because the major source of acetylcholine in the spinal cord is from sympathetic neurons (which are not present in our reduced preparation) we have no reason to think that there is very much acetylcholine signaling to inhibit. As a result, our expectation is that acetylcholine receptor antagonists would have no effect. In the case of substance P, although we know this peptide would be released in response to capsaicin, it is not the only important mediator that would be released into the cord — primary afferents also release glutamate and CGRP. Because TACR1 antagonists would only be inhibiting one of these three components (Substance P), it is

difficult to predict the degree to which TACR1 antagonists alone would inhibit activity in Ex1_{TACR3/1} neurons.

We acknowledge that it would be possible to address a slightly different question. In particular, the use of TACR1 receptor antagonists would allow us to test the role of substance P signaling in central sensitization. Although this is an interesting question, we feel that this experiment is beyond the scope of the current study for the following reasons:

- Substance P/Tacr1 signaling is used at the end of the experiment to define CICADA populations. We are concerned that, following prolonged inhibition of TACR1, we might no longer be able to visualize TACR1-mediated Ca²⁺ release. If the Ex1_{Tacr3/1}, Ex2_{TACR1}, and TACR1-expressing SPB neurons no longer respond appropriately to Substance P, we would no longer be able to identify these populations by CICADA, which would make our data uninterpretable. At a minimum, extensive control experiments would need to be performed to ensure that exposure to TACR1 antagonists does not alter CICADA profiling.
- One of the strengths of our study is the within-experiment design that allows us to compare the responses of the same cells before and after capsaicin treatment. This experimental design gives us very strong power to see specific changes across populations in the dorsal horn. In contrast, in an experiment in which we are comparing the responses observed across different preparations (i.e., capsaicin treatment in the presence of vehicle vs. capsaicin treatment in the presence of a TACR1 antagonist), our power to see changes in specific populations of neurons is much weaker. As a result, we anticipate that we would need to perform and analyze at least 12 more experiments (n = 6 per condition). Because it takes us about 1 month to analyze the data from each preparation, this experiment alone would be expected to take one year.

In light of the possible effects of TACR1 antagonists on CICADA profiling together with the enormous scope of the experiment, we humbly request that such an experiment is not necessary. In this regard, we note that there are already well-established roles for the necessity of TACR1-expressing cells for capsaicin induced sensitization [1-3]. These studies used SP-Saporin to deplete all TACR1 expressing cells (which would include Ex1_{TACR3/1}, Ex2_{TACR1} and TACR1-expressing SPB neurons) and found that TACR1-expressing neurons are necessary for capsaicin induced sensitization.

References:

1. Khasabov, S.G., et al., Spinal neurons that possess the substance P receptor are required for the development of central sensitization. J Neurosci, 2002. 22(20): p. 9086-98. DOI: 10.1523/JNEUROSCI.22-20-09086.2002
2. Mantyh, P.W., et al., Inhibition of hyperalgesia by ablation of lamina I spinal neurons expressing the substance P receptor. Science, 1997. 278(5336): p. 275-9. DOI: 10.1126/science.278.5336.275
3. Nichols, M.L., et al., Transmission of chronic nociception by spinal neurons expressing the substance P receptor. Science, 1999. 286(5444): p. 1558-61. DOI: 10.1126/science.286.5444.1558

2. One other minor point. The authors used von Frey filaments of 2.0 g for high threshold and 0.16 g for low threshold stimulation. For mice, at least with regards to hindpaw stimulation, high and low thresholds stimulation is generally much lower, 0.07 g and 0.4 g respectively. The skin prep the authors use incorporates the dorsal hind paw and proximal hip regions. Do these areas require greater von Frey stimulation to evoke a response or is it merely a consequence of the prep and the loss of underlying muscle and fascia?

These particular von Frey filaments were chosen as they represented a 'low' and 'high' threshold stimulation for behavioral experiments in mice rather than those which have been more commonly used for primary afferent studies. In our experience, mice have a 50% withdrawal threshold around 0.8 g [4] so we chose the 0.16 g filament as a 'low-threshold' filament as it would rarely produce responses in naïve animals (~5%), but after sensitization showed large increases in response. A 2.0 g filament was chosen as 'high-threshold' because it represented an effective maximum for behavioral responses. We acknowledge that these filaments (0.16 g and 2.0 g) are relatively high pressures compared to the mechanical *thresholds* of C-fibers. In prior work from our lab examining primary afferent response to cutaneous mechanical stimuli with von Frey filaments, the 50% response threshold for naïve C-fibers is around 0.04 grams (0.4 mN) with ~90% of C-fibers responding to a 0.1 gram (1 mN) stimulus [5]. However, these are strictly thresholds of response of the afferents, and we felt it best to examine pressures which produced behaviorally relevant responses within the context of the model of capsaicin induced sensitization.

References:

4. Warwick, C.A., et al., Mechanisms underlying mechanical sensitization induced by complement C5a: the roles of macrophages, TRPV1, and calcitonin gene-related peptide receptors. *Pain*, 2019. 160(3): p. 702-711. DOI: 10.1097/j.pain.0000000000001449
5. Albers, K.M., et al., Glial cell-line-derived neurotrophic factor expression in skin alters the mechanical sensitivity of cutaneous nociceptors. *J Neurosci*, 2006. 26(11): p. 2981-90. DOI: 10.1523/JNEUROSCI.4863-05.2006

Reviewer #2 (Remarks to the Author):

This paper mapped somatosensory receptive fields and modalities of spinoparabrachial (SPB) neurons and excitatory spinal interneurons in dorsal horn laminae I and 2 on a populational level by imaging Ca²⁺ activity using GCaMP6s and multiphoton microscopy both before and after capsaicin application to visualize central sensitization. This is a step towards understanding central sensitization with a long-term goal of understanding how, and what cellular mechanism central sensitization turns acute pain to chronic pain. Neurons are divided into classes by mapping specific Gq protein-coupled receptors in a process the authors named CICADA (Cell-type Identification by Ca²⁺-coupled Activity through Drug Activation). Neurons were categorized by lamina position, CICADA, sensory modality, and response to capsaicin application.

Strengths: Population level analysis of spinal cord somatosensory neurons is lacking to date and this paper represents a significant step forward in this area. Authors perform paired analysis of neurons before and after capsaicin application. Anesthesia confounds of these kinds of experiments are avoided. Neurons are classified by GPCR expression pattern, allowing researchers building on these results for clear cellular markers. Extended data shows a great deal of testing to determine limits and reproducibility of these methods.

Limitations: Authors acknowledge many limitations. Most somatosensory neurons were not labeled by CICADA which is limited to GPCRs that stimulate Ca²⁺ release from intracellular stores. Authors point out their method can be adapted to other forms of labeling that can detect different kinds of GPCRs. Imaging and analysis are labor and time intensive, limiting animal number. Ca²⁺ imaging limits analysis to superficial lamina of the dorsal horn, but

extended data suggests deeper imaging may be possible. It might be better authors develop this imaging method in vivo.

1. Given the most polymodal CICADA populations (Fig.4h) were also the most skewed towards lamina I (Fig.4f), I would like the authors to discuss more of the degree to which tuning can be predicted by CICADA category v. anatomical position. I.e., could the polymodality of Ex1, Ex2, and Ex3 be better predicted by almost entirely confined to lamina I? How does the tuning of unlabeled lamina I neurons compare to Ex1, Ex2, and Ex3?

This is an excellent point. We have included additional analyses to address this question in a new supplementary figure (Supplementary Fig 6d-f). To address this question, we utilized two approaches. First, we performed a multiple logistic regression where the outcome is MHC polymodality and we included depth and CICADA population as variables. We found that both variables (depth and CICADA cluster) were statistically predictive of MHC polymodality; however, we also found there was a significant interaction between depth and CICADA population which suggests neither variable can entirely predict the polymodality. To further address this question, we compared the Akaike's Information Criterion of the two regression models. We found that there is slightly less information lost in using CICADA population as a predictor versus depth (difference in Akaike's Information Criterion is only 0.014 which indicates that the models are not hugely different overall). Finally, we looked at the predictive power of these two models for MHC polymodality: 'CICADA population' provided a 73.9% negative and 48.5% positive predictive power, whereas 'depth' provides a 71.5% negative predictive power and no positive predictive power. Overall these analyses indicate that both depth and CICADA population contribute to the overall likelihood that a cell would be responsive to all three cardinal modalities (MHC polymodal), but that CICADA has an overall greater positive predictive power. These findings have been described in the result as follows: "*Notably, populations were significantly different from one another with respect to their tuning and response properties (Fig. 4h, Supplementary Fig. 6a-c), with populations in lamina I showing a greater degree of polymodality (Fig. 4i), even when comparing lamina I CICADA populations to depth-matched excitatory neurons that were not labeled by CICADA (Supplementary Fig. 6d-g).*" Additional information and statistical reporting has also been added to the methods and figure legends.

2. Authors analyzed lots of vector direction but [do] not explain why they do vector analysis and what each vector analysis means for results, central sensitization, or chronic pain, and sensitization. Would you please extensively articulate this for each vector analysis and figure?

We have now extended the rationale and interpretation of the vector analysis in more detail throughout the results and discussion sections to help the reader better understand and interpret these results.

3. Many experiments use very low or minimum animal number. Should increase at least n= 9 or higher.

We respectfully disagree with the reviewer about this point. The within-animal experimental design, in which we analyzed responses before and after capsaicin treatment, gave us very strong statistical power to see changes. Because all of the main endpoints of our study are statistically significant with n = 4 preparations, it is difficult to justify the need for more experiments. We also note that one of the authors, James Ibinson, is a statistician who

provided guidance for the appropriate statistical tests throughout the paper. Finally, we wish to point out that adding additional mice at this point would require us to re-analyze all of the data for many figures (i.e., Figures 5, 6, 7, 8 and Supplemental figures 7, 8 and 9.) Given that all findings of the paper are supported by appropriate statistical tests, we humbly ask the reviewer to reconsider this request.

4. Two animals showed large changes in low threshold receptive field sizes following capsaicin and two showed modest increases (Fig.6f). Are these differences a result of different numbers of neurons labeling in different lamina? I.e., did large changes correspond to animals having a higher percentage of labeled neurons in lamina I? A sentence or two in discussion and/or results could clarify this issue.

Thank you, we have added the following discussion, "*Of note, while changes in LT receptive field size post capsaicin are largest in Lamina II cells (Supplementary Fig. 8f-h) the per animal variability in the change post capsaicin was not well explained by a difference in average sampling depth and may represent another unknown source of variability.*"

Minor concerns

5. P18, line 413, second paragraph of Multiphoton Imaging section "filament was used to delineate a 15 x 15 mm region of skin was delineated that showed" Delete the crossed out words.

Thank you, this has been corrected

6. Figure2 label is wrong, Figure2i should be h.

Thank you, this has been corrected

Reviewer #3 (Remarks to the Author):

In this manuscript, Warwick and collaborators investigate the spinal dorsal horn network with calcium imaging to identify populations mediating capsaicin-induced central sensitization. To do so, they combine an ex vivo preparation that enables recording calcium activity at the cellular level with "natural" peripheral stimulation, with a pharmacological profiling strategy that enables a novel classification of dorsal horn neurons. Overall, the manuscript is very clear and well written, the experiments and their controls are adequately presented. However, it provides only limited new functional insights and omits discussing the abundant literature on dorsal horn networks, and on unmasking of low threshold inputs to neurons that normally respond to high threshold inputs (both in vitro and in vivo).

Main concerns:

1. The authors claim (l. 53) that only few studies have applied Ca²⁺ imaging to the spinal cord. They certainly refer to "in vivo" experiments, as there are countless in vitro calcium imaging studies. As for in vivo, citing only two of them is also misleading. From Johannssen & Helmchen's paper in JPhys 2010 (doi: 10.1113/jphysiol.2010.191833) to the in press Sullivan & Sdrulla one (JNeurosc., 2022, DOI: <https://doi.org/10.1523/JNEUROSCI.1860-21.2021>) many papers used this approach in vivo in anesthetized mice (e.g. Nishida et al. PLOS one, 2014, <https://doi.org/10.1371/journal.pone.0103321>).

We have rewritten the introduction to better represent these concerns: *“Ca²⁺ imaging is a powerful approach to visualize neural activity at a population level, and several groups have applied this approach to the spinal dorsal horn [6-13]. However, imaging studies in this region are challenging to perform due to respiratory movement and optical access [8] and difficult to interpret due to neuronal diversity [14].”*

References:

- Laffray, S., et al., *Adaptive movement compensation for in vivo imaging of fast cellular dynamics within a moving tissue*. PLoS One, 2011. 6(5): p. e19928.
- Johannssen, H.C. and F. Helmchen, *In vivo Ca²⁺ imaging of dorsal horn neuronal populations in mouse spinal cord*. J Physiol, 2010. 588(Pt 18): p. 3397-402.
- Johannssen, H.C. and F. Helmchen, *Two-photon imaging of spinal cord cellular networks*. Exp Neurol, 2013. 242: p. 18-26.
- Nishida, K., et al., *Three-dimensional distribution of sensory stimulation-evoked neuronal activity of spinal dorsal horn neurons analyzed by in vivo calcium imaging*. PLoS One, 2014. 9(8): p. e103321.
- Ran, C., M.A. Hoon, and X. Chen, *The coding of cutaneous temperature in the spinal cord*. Nat Neurosci, 2016. 19(9): p. 1201-9.
- Chisholm, K.I., et al., *Encoding of cutaneous stimuli by lamina I projection neurons*. Pain, 2021. 162(9): p. 2405-2417.
- Sullivan, S.J. and A.D. Sdrulla, *Excitatory and Inhibitory Neurons of the Spinal Cord Superficial Dorsal Horn Diverge in Their Somatosensory Responses and Plasticity in Vivo*. J Neurosci, 2022. 42(10): p. 1958-1973.
- Ran, C., G.N.A. Kamalani, and X. Chen, *Modality-Specific Modulation of Temperature Representations in the Spinal Cord after Injury*. J Neurosci, 2021. 41(39): p. 8210-8219.
- Todd, A.J., *Identifying functional populations among the interneurons in laminae I-III of the spinal dorsal horn*. Mol Pain, 2017. 13: p. 1744806917693003.

2. Interestingly, performing spinal cord calcium imaging in freely behaving mice has proven possible by the Nimmerjahn's lab in 2016 (NatCommun. 2016, DOI: 10.1038/ncomms11450). In contrast, the ex vivo preparation used here has limitation that should be discussed.

Although we agree that imaging spinal neurons in freely behaving animals using miniscopes is possible, we note that our particular study would not have been possible using mini-scopes because the yield is significantly lower than our method. In Sekiguchi *et al* (NatCommun. 2016, DOI: 10.1038/ncomms11450) [15] the authors use N=4 mice and analyze fewer than a hundred cells total, with <25 cells per animal. This small yield would preclude the types of analyses that was performed in this study. We now discuss the pros and cons of various imaging approaches as follows: *“We specifically chose an ex vivo approach because it offered several advantages over in vivo recordings: (1) it facilitated the development of CICADA, (2) provided higher quality recordings due to fewer movement artifacts, (3) more quantitative application of cutaneous stimuli, and (4) enabled recordings in the absence of anesthesia. Although this ex vivo approach has a number of appealing benefits, we acknowledge that it is a reduced preparation that lacks numerous components, notably descending modulation which has been well established to inhibit or facilitate nociceptive signaling. In addition, our ability to perform receptive field mapping is limited by the number of nerves and the size of skin attached, i.e. if a SDH neuron's receptive field would in vivo expand beyond the boundaries of the dissected piece of skin, we would not detect it and we are thus likely underestimating the degree of receptive field expansion.”*

3. Spinoparabrachial neurons have been previously studied with in vivo calcium imaging (Chisholm et al. Pain 2021) or in vivo electrophysiology (e.g. Keller et al., MolPain 2007, doi: 10.1186/1744-8069-3-27). There is no comparison of the present results with these

older studies, while for example the % of different cardinal modalities or polymodal neurons differs and should be discussed.

We agree that this type of comparison is beneficial and have included some comparisons. However, please note that it is difficult to compare our data directly to those in Chisholm *et al* [11] or Keller *et al* [16] due to differences in the cutaneous stimuli used. In particular, neither of these studies used comparable quantitative mechanical stimuli and the Keller study did not include any thermal stimuli. (Please note that we utilized calibrated von Frey filaments and we calculated the pressures applied using brush and airpuff stimuli in an effort to ensure that our experiments could be reproduced, and compared, in future studies.)

Based on the reviewers suggestion, we have now added the following sentences to the discussion *“Nevertheless, we see that our SPB neurons have broadly similar response properties with those measured in vivo [11, 16]. Examining mechanical stimuli specifically, we see that SPB neurons respond more to higher intensity stimulation, and after injury, increase their response to innocuous stimuli more than noxious stimuli [16] (Fig. 8b). In addition, we find that SPB neurons are predominantly polymodal (Fig. 2b-e), a finding similar to that reported by Chisholm et al., who performed Ca²⁺ imaging experiments in vivo and reported that 78% of SPB neurons are polymodal [11]”*

References:

11. Chisholm, K.I., et al., *Encoding of cutaneous stimuli by lamina I projection neurons*. Pain, 2021. 162(9): p. 2405-2417. DOI: 10.1097/j.pain.0000000000002226

16. Keller, A.F., et al., *Transformation of the output of spinal lamina I neurons after nerve injury and microglia stimulation underlying neuropathic pain*. Mol Pain, 2007. 3: p. 27. DOI: 10.1186/1744-8069-3-27

4. More generally, dorsal horn neurons have long been classified, either based on their electrophysiological profile, their morphology or more lately based on genetic markers. A significant contribution to the field would imply to validate the new profiling proposed here with at least one of the textbook classifications.

We agree and have included an extended discussion on this topic: *“This study represents the first attempt to categorize neurons based on their functional responses to GPCR ligands at a population level. A key remaining question is the degree to which we can relate the functional GPCR responses to which cells are predicted to show GPCR responses based on mRNA expression from single-cell sequencing studies. In some cases, there appears to be a strong correspondence between our functional data and the predictions from transcriptomics. For instance, comparing our putative populations to those described in Russ et al [17] suggests that Ex6_{CHRM3} may represent Excit-1, and that Ex5_{TRHR} may represent Excit-8. However, many gaps remain and future validation will be essential, particularly because mRNA expression does not perfectly predict functional responses of the corresponding proteins. Long term, the efforts to cross-reference cell subtypes that have been defined based on different experimental paradigms will be essential to create a uniform schema in the dorsal horn and enable new the modeling of somatosensory integration at a new level.”*

References:

17. Russ, D.E., et al., *A harmonized atlas of mouse spinal cord cell types and their spatial organization*. Nat Commun, 2021. 12(1): p. 5722. DOI: 10.1038/s41467-021-25125-1

5. This is even more true as the authors focus on capsaicine-induced plasticity, a subject that has been abundantly studied in vitro. Based on patch-clamp studies of capsaicine-responding neurons (directly or indirectly), several labs have proposed dorsal horn circuits models (among many others: Petitjean et al., EJN 2012, <https://doi.org/10.1111/j.1460-9568.2012.08273.x> ; Lu Y et al. J Clin Invest 2013, DOI: 10.1172/JCI70026 ; Ganley et al., JNeurosci 2015, DOI: 10.1523/JNEUROSCI.0406-15.2015). More recently, it was shown that the activation of calretinin neurons that are capsaicin responding and presynaptic to parabrachial neurons, induces allodynia and spontaneous pain (Petitjean et al., CellReports 2019, doi: 10.1016/j.celrep.2019.07.048. PMID: 31390558). Again, this literature should be cited and discussed, and attempts should be made to link the present results with what is already known on this circuit.

* The unmasking of low threshold inputs to neurons that normally respond only to high threshold inputs, as a substrate for allodynia, has long been discussed and demonstrated in the field. While some labs have focused on the disinhibition enabling this plasticity both in vitro or in vivo (Torsney & MacDermott, JNeurosci 2006, doi: 10.1523/JNEUROSCI.4584-05.2006 ; Lavertu et al., Brain 2014, <https://doi.org/10.1093/brain/awt334> ; Medrano et al., Pain 2016, DOI: 10.1097/j.pain.0000000000000538), others have investigated the underlying dorsal horn network: we mentioned above the implication of calretinin neurons (Petitjean et al., CellReports 2019) but NPY, PKCgamma or CCK neurons have also been implicated in neuropathic models (Tashima et al. PNAS 2021, DOI: 10.1073/pnas.2021220118 ; Peirs et al., Neuron 2021, DOI: 10.1016/j.neuron.2020.10.027). Importantly, the response profile of spinoparabrachial neurons has been shown to shift towards low threshold responses in a neuropathic pain model (Keller et al., MolPain 2007), while capsaicine was shown to markedly enhance the responses of excitatory neurons to innocuous touch (many neurons gaining sensitivity) (Sullivan & Sdrulla JNeurosc., 2022). A last general comment is that presenting only 20 references is difficult to justify considering all this valuable literature.

We agree and, within reasonable word limits, have included an extensively expanded introduction and discussion:

Introduction: “The spinal dorsal horn processes heterogeneous primary afferent input through complex local interneuron circuits. These neural circuits contribute to spinally mediated reflexes [18] and transmit nociceptive signals to a variety of supraspinal locations by way of spinal output neurons, particularly those in lamina I, most of which collateralize within the lateral parabrachial nucleus [18-22]. The large diversity of cell types within the superficial laminae have been defined with a variety of methods including morphology, location, intrinsic firing properties, expression profiling, and neurochemical markers [23] but we have yet to reach a consensus on the number and identity of bona fide cell types. Numerous studies have investigated the role of a given population for normal sensation and in the context of allodynia [17, 24-32]. A common thread in many of these studies is that any disruption in the balance between inhibition and excitation can drive allodynia via loss of inhibitory tone [18, 27, 28, 32-40] or sensitization of excitatory pathways [12, 41-44] including spinoparabrachial (SPB) neurons [1-3, 42, 45].”

Discussion: “Current models of neural circuitry in the dorsal horn suggest that after injury there are a variety of different mechanisms, many of which depend on the type of injury, that could

lead to an allodynia driven by central sensitization. One of the most established concepts is that allodynia is caused by disinhibition involving a loss of inhibitory neuron activity, which is especially common in the context of neuropathic injuries [18, 27, 28, 32-39]. Other studies have suggested sensitization of excitatory pathways through mechanisms such as long-term potentiation or alterations in intrinsic excitability [12, 41-44]. When comparing inhibitory and excitatory superficial neurons in vivo, only excitatory neurons had altered responses to low-threshold input after capsaicin [12]. This excitatory neuron specific enhancement could be driven by more ventral excitatory populations implicated in the capsaicin model of central sensitization including PKC γ cells present around the lamina II/III border, CCK cells in the deeper dorsal horn, and potentially calretinin neurons [25, 29]. Because our imaging was limited to the superficial dorsal horn (lamina I - II), it is unlikely that we imaged many, if any, PKC γ or CCK cells. These deeper populations could be providing low-threshold input to the superficial populations we examined here through disinhibition via deep dorsal horn populations [46] and parvalbumin inhibitory neurons [37] although it was not specifically examined in this work."

Full reference list at end of document.

Specific and/or minor points:

6. I. 48: have revealed "that" there are

Thank you. This has been changed.

7. I. 64: heterogeneous expression "of" Gq-coupled G-proteins

Thank you. This has been changed.

8. I. 89: document (here or in the Methods) the specificity of Cre expression in these mice

We have performed additional experiments including RNA-scope verification that the GCaMP6s expression within our mouse line is indeed exclusively excitatory, *i.e.* Vglut2 expressing. *"To verify that GCaMP6s was selectively expressed in excitatory neurons, we performed dual fluorescence in situ hybridization (FISH) using probes against Gcamp6s and Vglut2, and found that 97% of GCaMP6s neurons were positive for Vglut2 (Fig. 1d,e)."*

9. I. 128-130: The vector-based method is interesting to reduce the dimensionality of the data. However, its limitation should be discussed. In particular, a vector pointing to the HT direction could either reflect a specific HT neuron or a polymodal neuron with similar amplitude LT and heat response. As there are only 3 initial dimensions, one may wonder if a classical 3D representation would not be sufficient to visualize and analyze the data without loss of information.

We agree that the cardinal tuning in which we collapse all the mechanical and thermal information into a singular value for each modality has lost information and is a compromise for the sake of easier interpretation. In the context of a 3-dimensional representation, with X, Y, and Z dimensions there are still only 3 distinct dimensions to assign stimuli to (e.g. Mechanical, Heat, Cold). If you were to consider positive X and negative X as two separate stimuli, e.g. HT and LT as opposing vectors along the X-axis you could sum all the vectors in one graph. However, this becomes difficult to interpret: for example, consider a cell with equal HT and LT responses, it has no vector magnitude along the X axis suggesting a lack of response. A more sensitive approach would be to perform singular value decomposition (or another method) on this multi-dimensional dataset (*i.e.* Brush, HT, LT, airpuff, warm, cool, hot, and cold). This would

produce a lower dimensional dataset to be graphed/interpreted. However, this abstracts the data which limits direct interpretation as the resultant values no longer represent any of the individual variables completely. Each method has an advantage, and we felt that two separate sets rather than 1 abstract value, would be more interpretable to a broader audience and maintain some physiological relevance. We have expanded our discussion on vector analysis to help address these concerns: *“This way of visualizing the data is advantageous because it emphasizes the relative tuning of neurons to distinct types of stimuli. Because neurons in the superficial dorsal horn are rarely unimodal, we speculate that the sensory tuning may be the salient feature that is relevant for neural coding (Fig. 2). Although this method provides a new metric to quantify tuning, a limitation to this approach is that the absolute response amplitude to a given stimulus is obscured.”*

10. l. 145: include references for the somatotopy of DH organization

We have added the appropriate references, thank you.

11. l. 149: include here a brief description of receptive field size quantification approach

“Next, we examined how the intensity of stimulation affected receptive field size by stimulating at 16 sites equally spaced in a 4 x 4 grid using a HT, LT, or airpuff stimulus and summing the number of responses for each stimulation, averaged across 2 trials (Fig. 3d).”

12. l. 171 and following: indicate that the order of agonists application has been controlled for

“When applied in random order, eight of these ligands gave rise to robust Ca²⁺ transients....” Additionally, linear regression testing the order effects is also present in Supplementary Fig. 5e to test for rundown of responses in randomly applied drugs. There was no significant statistical trend between order of application (when randomized) and the amplitude of response.

13. l. 179: As mentioned in the Main concerns, the low % of CICADA responding neurons is an issue. Supplementary Fig. 5f shows that this is highly variable depending on the depth; this result should be in the main text and discussed.

“Overall, 25% of excitatory neurons responded to one or more CICADA ligands, although this proportion varied across laminae (32% in lamina I, 70% in lamina Ili, and 15% in lamina Ilo; Supplementary Fig. 5f).”

14. l. 185: cell clusters showed “qualitatively different” distributions according to depth; there is no statistical difference presented

An ordinary one-way ANOVA of the depth of the 7 CICADA populations showed a significant difference among means. Of 21 possible post-hoc multiple comparisons, 17 of 21 were significantly different in their mean depth and each comparison is now reported in Table 1.

“Moreover, cell clusters showed distinct distributions according to depth, consistent with the idea that they represent populations of bona fide cell types (Fig. 4f, Table 1).”

15. l. 191: Authors propose a classification of DH neurons according to their cardinal modalities (Fig. 2g); did they attempt to correlate this classification to the one emerging from CICADA (Ex1-7)?

We have included additional analyses and panels in the supplementary figures showing the cardinal and mechanical modalities of each of the CICADA populations: “*Notably, populations were significantly different from one another with respect to their tuning and response properties (Fig. 4h, Supplementary Fig. 6a-c)*”

16. I. 211: Ex1 was the only “CICADA” population that showed statistically significant predictive power for capsaicin response; the vast majority of excitatory neurons are not responding to CICADA. I. 215-216: again, among CICADA responding neurons, Ex1 is predominately driven, but other non-CICADA responding neurons may be strongly driven, this is not discussed.

We have rewritten this portion of the text: “*Of note, Ex1_{TACR3/1} was significantly enriched in capsaicin responders relative to other lamina I cells (44% of Ex1_{TACR3/1} vs 8% of depth-matched excitatory neurons that were not labeled by CICADA ligands) even though Ex1_{TACR3/1} makes up <10% of excitatory neurons in lamina I. Thus, intradermal capsaicin drives activity in excitatory neurons in the superficial dorsal horn, and this activity is significantly enriched within Ex1_{TACR3/1}, a population of relatively untuned (Fig. 5k-l), polymodal neurons (Fig. 5j) that are located in lamina I.*”

17. I. 220: These long experiments raise two questions: * what is the stability of cutaneous stimulation responses without capsaicin treatment?

These *ex vivo* preparations remain useful for experimentation ~6 hours [47]. The described capsaicin experiments last 3-4 hours and so reside well within an expected window of health.

18. * CICADA is performed at the end: could possibly the application of capsaicin induce a plasticity in the CICADA profile?

We acknowledge that it is theoretically possible for capsaicin treatment to change the receptor expression; however, within the relatively short time frame post capsaicin (1-2 hours) we do not expect protein degradation/expression to change the profile of receptors in the membrane dramatically. This view is supported by comparisons of CICADA profiles in capsaicin-treated experiments to CICADA profiles in initial experiments when we were developing the technique and screening agonists (in the absence of capsaicin). Because the proportions of neurons that responded to each CICADA agonist was not different under these two conditions, it is unlikely that capsaicin is inducing plasticity in the CICADA profile

19. I. 222: remove “to LT stimulate” (said in I. 221)

Thank you. This has been changed.

20. I. 251-260: A striking observation for the plasticity of Ex2 is also the fact that they are less responsive to Heat after capsaicin according to Fig. 7g. This should be presented in the results and discussed.

We agree that is an unexpected observation that we do not yet totally understand. Based on the reviewer’s suggestion, we now highlight this observation in the text, as follows.

Results: “*Following capsaicin treatment, however, Ex2_{TACR1} showed a significant change in tuning towards LT input, indicating a large amplification of LT stimuli (Fig. 7h) as well an increase in the percent of cells responsive to LT stimuli and an intriguing reduction in heat responsiveness (Fig. 7g).*” **Discussion:** “*However, using this large Peltier ensures we have*

covered the entirety of the skin's receptive zone, which makes the decreased response to heat after capsaicin induced sensitization within Ex2_{TACR1} (Fig. 7g) as well as the reduction in amplitudes of response to heat (Supplementary Fig. 8d) particularly interesting. As Ex2_{TACR1} is basically the most heat tuned population (Fig. 4h) and is the only CICADA population with decreases in heat responsiveness (Fig. 7), this suggests that capsaicin-induced central sensitization affects multiple modalities to shape sensory tuning rather than simply enhancing mechanical pathways."

21. I. 548: Why is the definition of capsaicin responsive neurons so different from CICADA?

CICADA responses are quantified in the presence of TTX which prevents action potential formation, which means the Ca²⁺ release caused by a CICADA ligand is derived from the endoplasmic reticulum which causes a singular peak which promptly resolves as the cell cannot produce additional AP-dependent increases in Ca²⁺ due to the TTX. In contrast, the Ca²⁺ responses after the injection of capsaicin are (presumably) due to a mixture of AP-dependent, glutamate receptor, and neuropeptide receptor dependent Ca²⁺ release. These varied sources of Ca²⁺ lead to non-homogenous Ca²⁺ traces in addition to an extended time frame for activation, e.g. some cells respond immediately to the injection and quiet down with seconds whereas others show significantly increased activity for minutes with occasional seconds of quiescence. This led to the approach in the paper which requires an average DF/F of 150% of pre-capsaicin levels based on a 2-min average. This 2-minute time frame and threshold permits a singular metric which captures the heterogeneous mix of activity post capsaicin.

22. Fig. 2: where is 2g? In 2i, the "brush static LT" population is mislabeled N=1265 cells but how many SPB?

Thank you. This has been fixed and N for SPN neurons has been added to the figure legend (22).

23. Fig. 3: Indicate number of neurons in 3e, indicate briefly how RF size is quantified

This has been included in the figure legend.

24. How is 3h built? It does not look like the RF size/depth of all of the neurons that would look more like a points cloud. Is it an average RF size for a certain depth interval? The variation should be presented if this is the case.

It is indeed a smoothed average as described in the methods "For all data plotted as a function of depth (e.g. Fig. 3h) smoothing was applied using Prism (9 neighbor, 6th order polynomial). The graph has been updated to include non-smoothed individual data points to better present the variation.

25. 3i: again, if each point represents a mouse, authors should illustrate mean and dispersion.

The mean and SEM has now been included.

26. Fig. 4: Is the color code in 4b the one used for the rest of the figure? Is it the same cell represented with a given color in all of 4b panels? The blue cell per example (Ex1??) has a strong response to Taltirelin which does not fit with 4d.

It is not the same cell throughout Fig. 4b despite the color scheme matching the CICADA labels. Each trace is a different cell for Fig. 4b. We have updated the color scheme to make it more obvious that these are individual cells.

27. In 4h, why use two tests to compare the vector angle and magnitude? A two-way ANOVA could be used to take into account these two aspects of the vector.

As vector magnitude and angle are two metrics taken from the same vector, these two aspects are not independent of each other, as required for a 2-way ANOVA, hence the 2 separate one-way ANOVAs.

28. Fig. 5: A scale is missing in 5f for RC/LM distance

Thank you, this has been fixed.

29. Fig. 6: N=4 mice in 6e?

Yes.

30. In 6f one expects an average and dispersion for the RF size by animal.

Mean and SEM has been added to Fig. 6e and f.

31. Fig. 8: Why use the “old” terminology in 8c while they have presented another new and quantitative way of classifying neurons according to their cardinal modality in 2f?

We agree that the old categorization scheme (LT, WDR, and HT) is too simplistic to capture the diversity of spinal output neurons. However, we do not yet have enough data create a better classification scheme. The use of LT, WDR, and HT was used in Fig. 8 because there is an abundance of literature already using these classifications of spinal projection neurons and we felt it would be beneficial to the field to present our data in the same manner as prior studies. This was to show both validation of Ca^{2+} imaging against electrophysiological datasets as well as providing a more comparable point of reference.

32. How is the distribution in 8c modified after capsaicin?

For the purposes of Fig. 8c and d, the SPB neurons were categorized using baseline responses and not re-assessed post-capsaicin so that the groups would remain consistent. Examining the post-capsaicin responses, the number of cells which responded to LT stimuli did not change (unlike the interneurons) and nearly all of the changes in the vectors originate primarily from changes in amplitude of response for both LT and HT responses.

33. What is the “max response” used to normalize the data in 8e?

It is the same metric used in Fig. 4e: for each cell, the response to each ligand was normalized to the largest response of any CICADA ligand, e.g. if the largest amplitude of response was 350% to SP, then all values for that cell are normalized against SP. These values then range from 0 to 100% for each cell and then the population is averaged and plotted.

34. Supplementary Fig. 4a: Top left panel presents red only, so the green labeled is not needed.

Thank you, we have corrected this.

35. Middle left panel [Supplementary Fig. 4a]: what stands the left panel “ventral” for?

It should not be there, and is likely to have been accidentally copied from the orthogonal view panel below. Thank you for catching this mistake.

36. Legend of panel g,h [Supplementary Fig. 4a]: for the preference score, why divide by X_{\max} and not X_{avg} ?

X_{\max} was chosen over X_{avg} so that the values would lie within a range of -1 to +1. If X_{avg} is used, the minimum value is -1 but the maximum is not. e.g. values of 3, 0, 0, result in a preference score of 2, -1, -1 for x_{avg} but 0.66, -0.33, -0.33 for X_{\max} and thus creating a preference value skewed for positive values.

37. Panel I [Supplementary Fig. 4]: only the most superficial cells showing “some” tuning for heat; their response is most tuned for mechanical stimulations.

We agree, mechanical responsiveness is generally pervasive throughout the spinal cord and makes up the most common modality of responsiveness by far. This is why it even the superficial population as whole (on average) is mechanically tuned. This is certainly not to say that all cells are uniformly mechanically tuned, but these graphs represent general trends rather than a clear rule.

38. Legend of panel j (“In particular, the most superficial cells are more tuned for heat, Lam Ilo cells tend to have high amplitude responses (high magnitude vector), whereas Lam Ili begins to show smaller magnitude mechanical responses relative to other modalities all while following the general trend of preferring mechanical stimuli at deeper lamina”) looks like a strong assertion but lacks statistical analysis.

We agree. These statements are descriptive of the data and were intended to be used to assist the reader while examining this novel type of graph rather than making a strong declaration per se. They have been reworded to a more neutral description.

1. Khasabov, S.G., et al., *Spinal neurons that possess the substance P receptor are required for the development of central sensitization*. J Neurosci, 2002. **22**(20): p. 9086-98.
2. Mantyh, P.W., et al., *Inhibition of hyperalgesia by ablation of lamina I spinal neurons expressing the substance P receptor*. Science, 1997. **278**(5336): p. 275-9.
3. Nichols, M.L., et al., *Transmission of chronic nociception by spinal neurons expressing the substance P receptor*. Science, 1999. **286**(5444): p. 1558-61.
4. Warwick, C.A., et al., *Mechanisms underlying mechanical sensitization induced by complement C5a: the roles of macrophages, TRPV1, and calcitonin gene-related peptide receptors*. Pain, 2019. **160**(3): p. 702-711.
5. Albers, K.M., et al., *Glial cell-line-derived neurotrophic factor expression in skin alters the mechanical sensitivity of cutaneous nociceptors*. J Neurosci, 2006. **26**(11): p. 2981-90.
6. Laffray, S., et al., *Adaptive movement compensation for in vivo imaging of fast cellular dynamics within a moving tissue*. PLoS One, 2011. **6**(5): p. e19928.
7. Johannssen, H.C. and F. Helmchen, *In vivo Ca²⁺ imaging of dorsal horn neuronal populations in mouse spinal cord*. J Physiol, 2010. **588**(Pt 18): p. 3397-402.
8. Johannssen, H.C. and F. Helmchen, *Two-photon imaging of spinal cord cellular networks*. Exp Neurol, 2013. **242**: p. 18-26.

9. Nishida, K., et al., *Three-dimensional distribution of sensory stimulation-evoked neuronal activity of spinal dorsal horn neurons analyzed by in vivo calcium imaging*. PLoS One, 2014. **9**(8): p. e103321.
10. Ran, C., M.A. Hoon, and X. Chen, *The coding of cutaneous temperature in the spinal cord*. Nat Neurosci, 2016. **19**(9): p. 1201-9.
11. Chisholm, K.I., et al., *Encoding of cutaneous stimuli by lamina I projection neurons*. Pain, 2021. **162**(9): p. 2405-2417.
12. Sullivan, S.J. and A.D. Sdrulla, *Excitatory and Inhibitory Neurons of the Spinal Cord Superficial Dorsal Horn Diverge in Their Somatosensory Responses and Plasticity in Vivo*. J Neurosci, 2022. **42**(10): p. 1958-1973.
13. Ran, C., G.N.A. Kamalani, and X. Chen, *Modality-Specific Modulation of Temperature Representations in the Spinal Cord after Injury*. J Neurosci, 2021. **41**(39): p. 8210-8219.
14. Todd, A.J., *Identifying functional populations among the interneurons in laminae I-III of the spinal dorsal horn*. Mol Pain, 2017. **13**: p. 1744806917693003.
15. Sekiguchi, K.J., et al., *Imaging large-scale cellular activity in spinal cord of freely behaving mice*. Nat Commun, 2016. **7**: p. 11450.
16. Keller, A.F., et al., *Transformation of the output of spinal lamina I neurons after nerve injury and microglia stimulation underlying neuropathic pain*. Mol Pain, 2007. **3**: p. 27.
17. Russ, D.E., et al., *A harmonized atlas of mouse spinal cord cell types and their spatial organization*. Nat Commun, 2021. **12**(1): p. 5722.
18. Sivilotti, L. and C.J. Woolf, *The contribution of GABAA and glycine receptors to central sensitization: disinhibition and touch-evoked allodynia in the spinal cord*. J Neurophysiol, 1994. **72**(1): p. 169-79.
19. Todd, A.J., *Neuronal circuitry for pain processing in the dorsal horn*. Nat Rev Neurosci, 2010. **11**(12): p. 823-36.
20. Todd, A.J., et al., *Projection neurons in lamina I of rat spinal cord with the neurokinin 1 receptor are selectively innervated by substance p-containing afferents and respond to noxious stimulation*. J Neurosci, 2002. **22**(10): p. 4103-13.
21. Todd, A.J., et al., *Fos induction in lamina I projection neurons in response to noxious thermal stimuli*. Neuroscience, 2005. **131**(1): p. 209-17.
22. Cameron, D., et al., *The organisation of spinoparabrachial neurons in the mouse*. Pain, 2015. **156**(10): p. 2061-2071.
23. Graham, B.A. and D.I. Hughes, *Defining populations of dorsal horn interneurons*. Pain, 2020. **161**(11): p. 2434-2436.
24. Peirs, C., R. Dallel, and A.J. Todd, *Recent advances in our understanding of the organization of dorsal horn neuron populations and their contribution to cutaneous mechanical allodynia*. J Neural Transm (Vienna), 2020. **127**(4): p. 505-525.
25. Peirs, C., et al., *Mechanical Allodynia Circuitry in the Dorsal Horn Is Defined by the Nature of the Injury*. Neuron, 2021. **109**(1): p. 73-90 e7.
26. Moehring, F., et al., *Uncovering the Cells and Circuits of Touch in Normal and Pathological Settings*. Neuron, 2018. **100**(2): p. 349-360.
27. Bourane, S., et al., *Gate control of mechanical itch by a subpopulation of spinal cord interneurons*. Science, 2015. **350**(6260): p. 550-4.
28. Lu, Y., et al., *A feed-forward spinal cord glycinergic neural circuit gates mechanical allodynia*. J Clin Invest, 2013. **123**(9): p. 4050-62.
29. Petitjean, H., et al., *Recruitment of Spinoparabrachial Neurons by Dorsal Horn Calretinin Neurons*. Cell Rep, 2019. **28**(6): p. 1429-1438 e4.

30. Smith, K.M., et al., *Calretinin positive neurons form an excitatory amplifier network in the spinal cord dorsal horn*. *Elife*, 2019. **8**.
31. Ganley, R.P., et al., *Inhibitory Interneurons That Express GFP in the PrP-GFP Mouse Spinal Cord Are Morphologically Heterogeneous, Innervated by Several Classes of Primary Afferent and Include Lamina I Projection Neurons among Their Postsynaptic Targets*. *J Neurosci*, 2015. **35**(19): p. 7626-42.
32. Tashima, R., et al., *A subset of spinal dorsal horn interneurons crucial for gating touch-evoked pain-like behavior*. *Proc Natl Acad Sci U S A*, 2021. **118**(3).
33. Yaksh, T.L., *Behavioral and autonomic correlates of the tactile evoked allodynia produced by spinal glycine inhibition: effects of modulatory receptor systems and excitatory amino acid antagonists*. *Pain*, 1989. **37**(1): p. 111-123.
34. Miraucourt, L.S., R. Dallel, and D.L. Voisin, *Glycine inhibitory dysfunction turns touch into pain through PKCgamma interneurons*. *PLoS One*, 2007. **2**(11): p. e1116.
35. Torsney, C. and A.B. MacDermott, *Disinhibition opens the gate to pathological pain signaling in superficial neurokinin 1 receptor-expressing neurons in rat spinal cord*. *J Neurosci*, 2006. **26**(6): p. 1833-43.
36. Medrano, M.C., et al., *Loss of inhibitory tone on spinal cord dorsal horn spontaneously and nonspontaneously active neurons in a mouse model of neuropathic pain*. *Pain*, 2016. **157**(7): p. 1432-1442.
37. Petitjean, H., et al., *Dorsal Horn Parvalbumin Neurons Are Gate-Keepers of Touch-Evoked Pain after Nerve Injury*. *Cell Rep*, 2015. **13**(6): p. 1246-1257.
38. Hwang, J.H. and T.L. Yaksh, *The effect of spinal GABA receptor agonists on tactile allodynia in a surgically-induced neuropathic pain model in the rat*. *Pain*, 1997. **70**(1): p. 15-22.
39. Malan, T.P., H.P. Mata, and F. Porreca, *Spinal GABA(A) and GABA(B) receptor pharmacology in a rat model of neuropathic pain*. *Anesthesiology*, 2002. **96**(5): p. 1161-7.
40. Zeilhofer, H.U., H. Wildner, and G.E. Yevenes, *Fast synaptic inhibition in spinal sensory processing and pain control*. *Physiol Rev*, 2012. **92**(1): p. 193-235.
41. Lavertu, G., S.L. Cote, and Y. De Koninck, *Enhancing K-Cl co-transport restores normal spinothalamic sensory coding in a neuropathic pain model*. *Brain*, 2014. **137**(Pt 3): p. 724-38.
42. Ikeda, H., et al., *Synaptic plasticity in spinal lamina I projection neurons that mediate hyperalgesia*. *Science*, 2003. **299**(5610): p. 1237-40.
43. Ikeda, H., et al., *Synaptic amplifier of inflammatory pain in the spinal dorsal horn*. *Science*, 2006. **312**(5780): p. 1659-62.
44. Hu, H.J., et al., *The kv4.2 potassium channel subunit is required for pain plasticity*. *Neuron*, 2006. **50**(1): p. 89-100.
45. Suzuki, R., et al., *Superficial NK1-expressing neurons control spinal excitability through activation of descending pathways*. *Nat Neurosci*, 2002. **5**(12): p. 1319-26.
46. Petitjean, H., J.L. Rodeau, and R. Schlichter, *Interactions between superficial and deep dorsal horn spinal cord neurons in the processing of nociceptive information*. *Eur J Neurosci*, 2012. **36**(11): p. 3500-8.
47. Hachisuka, J., et al., *Semi-intact ex vivo approach to investigate spinal somatosensory circuits*. *Elife*, 2016. **5**.

REVIEWERS' COMMENTS

Reviewer #1 (Remarks to the Author):

The authors response to my concerns and questions are sufficient. Explanation as to why the proposed experiments are out of the scope of this report are well taken. This reviewer appreciates the effort the authors take in explanations and agrees that the use of TACR1 receptor antagonists is out of the scope of this paper.

Reviewer #2 (Remarks to the Author):

The authors have addressed my previous comments.
With the revision of the manuscript, the significance of this study has been improved.

Reviewer #3 (Remarks to the Author):

Warwick and collaborators have considerably revised their manuscript to address the reviewers concerns. The results are now better put in context with previous studies, giving a fair acknowledgment of what represents new insights vs. confirmation of previous data with new approaches. As mentioned in the first review, the experiments and controls are adequately performed and presented, and are now better discussed. Some minor concerns remain that are listed below:

1) (response point 2) Advantages of CICADA as presented in the intro: your response mentioned an interesting point that is not explicitly mentioned in the Introduction, the large yield vs. miniscope studies (I guess it is in the first point, but it could be more explicit)

2) (response point 3) Authors have included some discussion of previous recording of SPB neurons, but say in their response that they were limited by the differences in the cutaneous stimuli used. Yet all of them present brush stimuli, for example, with 0% response in Chisholm (mouse, urethane), 21% in Keller (rat, pentobarbital) and 53% of responses in the present study. As for high threshold mechanical stimuli, the 2.0g VFrey filament (100% responses in the present study) could still be discussed in light with pinch responses: 42% in Chisholm, 100% in Keller. As for thermal stimuli, 92% of SPB neurons respond to cold in Chisholm, vs. 47% in the present study, but 85% in your previous study with electrophysiological recordings in the same prep (Elife 2016) etc. Bester et al. (JNeurophys 2000) also provides ephys recording of these neurons in anesthetized rats.

There are therefore some differences that can certainly be explained by large by the prep difference. Yet the sentence "our SPB neurons have broadly similar response properties with those measured in vivo" is slightly misleading and authors could go a bit more into the details.

3) (response point 3) The following sentence "we see that SPB neurons respond more to higher intensity stimulation..." is also misleading, as the reader might understand that these represent your own results. "Keller et al. see that SPB..." would be more appropriate.

4) (response point 4) Try to rephrase the sentence ". A key remaining question is the degree to which we can relate the functional GPCR responses to which cells are predicted to show GPCR responses based on mRNA expression from single-cell sequencing studies." which is really difficult to understand.

5) (response point 5) Reference to Chisholm and Keller's papers could be included at the end of the intro paragraph you propose.

6) (response point 9) My proposal was not to analyze concomitantly the 6 stimuli, which would indeed require to reduce the multi-dimensional dataset, but in line with the 2-step analysis proposed by the authors to represent the data in 3D (as there are only 3 parameters for each step) rather than with the vector-based method. A point in 3D has 3 coordinates, so we maintain all of the original information, while with the length and angle of the vector, we cannot go back to

the data. 3D data is still "readable" by the human eye. However we agree that the limit of the present approach is now explicitly mentioned in the manuscript.

7) (response point 11) Maybe mention explicitly that you count the number of responding sites: "and summing the number of RESPONSIVE SITES for each stimulation, averaged across 2 trials (Fig. 3d)."

8) (response point 17) I could not find the mentioned stability of cutaneous stimulation responses over hours in the 2016 Elife article. The question is not so much the "usability" of the prep but really the stability of responses over repetitive stimulations. The authors may have some trace to illustrate that?

9) (response point 18) The comparison with their initial experiments without capsaicin answers the point and it would be great to include it as a suppl data if traces are available.

10) (response point 33) In Fig. 4e, it is indicated that the responses are normalized to the max response of the indicated ligand, not to any ligand as mentioned in the response for fig. 8e. It might be worth presenting this representation in the methods so that it clearer for the reader.

11) (response point 37) The initial proposition was to rephrase the legend of panel i, as the most superficial cells don't really show a "significant tuning for heat", but rather show relatively more tuning for heat than in other laminae

12) (manuscript l. 208-210) While Fig. 4i directly presents the degree of polymodality, this does not appear explicitly in suppl. Fig. 6d-g. The sentence might need rephrasing.

13) (manuscript l. 310) It should be specified that the manuscript presents data on "excitatory" neurons of the "superficial" dorsal horn.

14) (manuscript l. 315, 327) The word "pain" should be carefully used if referring to animal studies.

15) (manuscript l. 322) Subtitle requires rephrasing, authors do not demonstrate that the Ex1 population initiates central sensitization.

16) (manuscript l. 673) Probably transverse sections?

17) (suppl. Fig. 6) Panel e presents cardinal tuning and mentions that they are compared to the AVERAGE of all non-labelled lamina I cells? Doesn't the comparison take into account all of the individual points?

The legend also indicates that the vector angle and magnitude are compared with the AVERAGE of ALL lamina I cells (not individual points? Not only of lamina I cells not-labelled with CICADA?) accepted for publication.

Blue text indicates the authors' response to reviewers

Red text indicates additions or changes in the text of manuscript

Reviewer #1 (Remarks to the Author):

The authors response to my concerns and questions are sufficient. Explanation as to why the proposed experiments are out of the scope of this report are well taken. This reviewer appreciates the effort the authors take in explanations and agrees that the use of TACR1 receptor antagonists is out of the scope of this paper.

Reviewer #2 (Remarks to the Author):

The authors have addressed my previous comments.
With the revision of the manuscript, the significance of this study has been improved.

Reviewer #3 (Remarks to the Author):

Warwick and collaborators have considerably revised their manuscript to address the reviewers concerns. The results are now better put in context with previous studies, giving a fair acknowledgment of what represents new insights vs. confirmation of previous data with new approaches. As mentioned in the first review, the experiments and controls are adequately performed and presented, and are now better discussed. Some minor concerns remain that are listed below:

1. ([1st revision] response point 2) Advantages of CICADA as presented in the intro: your response mentioned an interesting point that is not explicitly mentioned in the Introduction, the large yield vs. miniscope studies (I guess it is in the first point, but it could be more explicit)

We have additionally revised our discussion on the techniques in the following paragraph:

“We specifically chose an *ex vivo* approach as it offered several advantages over *in vivo* recordings because it: (1) facilitated the development of CICADA, (2) yielded hundreds of neurons per animal, which is required for population-scale analysis, (3) provided higher quality recordings due to fewer movement artifacts, (4) facilitated a more quantitative application of cutaneous stimuli, and (5) enabled recordings in the absence of anesthesia.”

2. (Response point 3) Authors have included some discussion of previous recording of SPB neurons, but say in their response that they were limited by the differences in the cutaneous stimuli used. Yet all of them present brush stimuli, for example, with 0% response in Chisholm (mouse, urethane), 21% in Keller (rat, pentobarbital) and 53% of responses in the present study. As for high threshold mechanical stimuli, the 2.0g VFrey filament (100% responses in the present study) could still be discussed in light with pinch responses: 42% in Chisholm, 100% in Keller. As for thermal stimuli, 92% of SPB neurons respond to cold in Chisholm, vs. 47% in the present study, but 85% in your previous study with electrophysiological recordings in the same prep (Elife 2016) etc.

Bester et al. (JNeurophys 2000) also provides ephys recording of these neurons in anesthetized rats. There are therefore some differences that can certainly be explained by large by the prep difference. Yet the sentence “our SPB neurons

have broadly similar response properties with those measured in vivo” is slightly misleading and authors could go a bit more into the details.

We have included additional discussion about the response properties of SPB neurons across a number of relevant studies. We have also added a new table to summarize these studies and a new panel comparing our results to others in supplementary fig. 4.

“Nevertheless, we see that our SPB neurons have broadly similar response properties with those measured in vivo (Supplementary Table 2) [35, 49, 58-62]. However, it should be noted that SPB neurons generally have heterogeneous response properties, and there is significant variability across previously published studies. These inconsistencies are likely due to a variety of methodological differences such as sampling biases, anesthesia, species differences, and, to some extent, the use of qualitative stimuli such as brushing and pinching (Supplementary Table 2). If we compare our study to those that reported both force and area for their static mechanical stimuli, we find our data aligns quite well with in vivo recordings (Supplementary Fig. 4f). We observed that SPB neurons from naïve mice respond in graded fashion across mechanical intensity (Fig. 8b), consistent with prior in vivo studies [60, 61]. We likewise found that, although many changes are observed following injury, the responses to innocuous stimuli show the greatest degree of change similar to prior reports [35]. Finally, we find that SPB neurons are predominantly polymodal (Fig. 2b-e), a finding consistent with existing literature, where reported polymodality in SPB neurons has ranged from 34% to 93% (average of 59%) (Supplementary Table 2) [35, 49, 58-61, 63, 64].”

Author	PMID	Species	Method	Anesthesia or Ex vivo	% Brush or Light Touch	Brush Pressure	% Pinch	Pinch Pressure	Mech	Heat	Cold	MHC	Any polymodality	Thermal Method
Han	10195146	Cat	Intracellular Recording	Pentobarbital	Data not shown	Not quantified	66%	Not quantified	66%	45%	52%	18%	45%	Water jet
Light	8237218	Cat	Intracellular Recording	Pentobarbital	9%	Not quantified	79%	Not quantified	88%	23%	23%	0%	34%	Thermode
Warwick		Mouse	Multiphoton Ca2+ Imaging	Ex vivo	53%	6.5 kPa	--	Not Applied	100%	52%	47%	48%	52%	Thermode
Hachisuka	27991851 + 31577643	Mouse	Patch Clamp Recording	Ex vivo	Data not shown	Not quantified	--	Not Applied	70%	33%	59%	43%	41%	Water jet
Allard	30719699	Mouse	Extracellular Recording	Isoflurane	23.21%	Not quantified	100%	2370 mN (no area)	83%	83%	64%	~50%*	80%	Water jet
Chisholm	33769365	Mouse	Epifluorescence Ca2+ Imaging	Urethane	0%	Not quantified	42%	Not quantified	72%	25%	85%	18%	70%	Thermode
Bester	10758132	Rat	Extracellular Recording	Halothane & nitrous oxide	25%	Not quantified	92%	Not quantified	92%	100%	32%	35%	92%	Water jet
Keller	17900333	Rat	Extracellular Recording	Pentobarbital or ketamine/xylazine	21%	Not quantified	100%	1250 kPa (100g/1mm)	100%	N/A	N/A	N/A	N/A	N/A

3. (Response point 3) The following sentence “we see that SPB neurons respond more to higher intensity stimulation...” is also misleading, as the reader might understand that these represent your own results. “Keller et al. see that SPB...” would be more appropriate.

We have reworded the statements to ensure each conclusion is appropriately attributed. Of note, we do see that SPB neurons respond with larger amplitude responses to high intensity stimuli as shown in Fig. 8b consistent with many prior studies that many SPB neurons encode the intensity of the mechanical stimuli.

“We observed that SPB neurons from naïve mice respond in graded fashion across mechanical intensity (Fig. 8b), consistent with prior in vivo studies [60, 61]. We likewise found that, although many changes are observed following injury, the responses to innocuous stimuli show the greatest degree of change similar to prior reports [35]. Finally, we find that SPB neurons are predominantly polymodal (Fig. 2b-e), a finding consistent with existing literature, where reported polymodality in SPB neurons has ranged from 34% to 93% (average of 59%) (Supplementary Table 2) [35, 49, 58-61, 63, 64].”

4. (Response point 4) Try to rephrase the sentence “. A key remaining question is the degree to which we can relate the functional GPCR responses to which cells are predicted to show GPCR responses based on mRNA expression from single-cell sequencing studies.” which is really difficult to understand.

We have reworded this sentence to hopefully clarify our intent:

“A key remaining question is how well can we relate our results to single-cell sequencing studies. In particular, we do not yet know to what degree the expression of mRNA will predict the functional GPCR responses we have observed here.”

5. (Response point 5) Reference to Chisholm and Keller’s papers could be included at the end of the intro paragraph you propose.

Thank you, we have added the relevant references.

6. (Response point 9) My proposal was not to analyze concomitantly the 6 stimuli, which would indeed require to reduce the multi-dimensional dataset, but in line with the 2-step analysis proposed by the authors to represent the data in 3D (as there are only 3 parameters for each step) rather than with the vector-based method. A point in 3D has 3 coordinates, so we maintain all of the original information, while with the length and angle of the vector, we cannot go back to the data. 3D data is still “readable” by the human eye. However, we agree that the limit of the present approach is now explicitly mentioned in the manuscript.

Thank you for clarifying. We agree that presenting cardinal and mechanical stimuli separately with a 3D axis would be possible.

7. (Response point 11) Maybe mention explicitly that you count the number of responding sites:

“and summing the number of RESPONSIVE SITES for each stimulation, averaged across 2 trials (Fig. 3d).”

This has been clarified in the text as follows.

“and summing the number of responsive sites for each stimulation”

8. (Response point 17) I could not find the mentioned stability of cutaneous stimulation responses over hours in the 2016 Elife article. The question is not so much the “usability” of the prep but really the stability of responses over repetitive stimulations. The authors may have some trace to illustrate that?

Within the 2016 eLife article the relevant text was as follows: “Thereafter, recordings can be performed for up to 6 hr post-dissection.” Although there was no data presented in support of this assertion in the eLife article, we show in Supplementary Figure 2 that for repeated brushing, linear regression analysis of brush responses shows no correlation between the brush trial and the amplitude of response. The traces shown in Supplementary Fig. 2d brushed the skin 5-6 times every 30 seconds, then we waited for an hour and then repeated the brushing. These data suggested that within a short period of repeated stimulation there was minimal rundown and that over a longer time scale (1 hour in this case) the responses remained stable.

9. (Response point 18) The comparison with their initial experiments without capsaicin answers the point and it would be great to include it as a suppl data if traces are available.

We have added an additional panel into Supplementary Fig. 9e showing the percent responders in Naïve and Capsaicin treated animals for CICADA ligands which we have matching data for, i.e., it used the same dose as the final version of CICADA shown in this paper. While these are post-hoc analyses and we did not power them for this non-repeated measures comparison, there were no statistically significant differences or consistent trends.

“Percent of CICADA labeled cells responsive to the indicated CICADA ligands averaged by animal in either naïve preparations or animals which were subjected to the cutaneous stimulation protocol and capsaicin injection described in Fig. 6b (caps). No significant differences between naïve and capsaicin-treated preparations were found.”

10. (Response point 33) In Fig. 4e, it is indicated that the responses are normalized to the max response of the indicated ligand, not to any ligand as mentioned in the response for fig. 8e. It might be worth presenting this representation in the methods so that it clearer for the reader

We have added a brief explanation in the figure legend as well as a more verbose explanation in the methods:

“Radar charts showing the % maximum response for CICADA ligands, as in Fig. 4e and Fig. 8e are min to max normalized responses of responsive cells. For cells which responded to at least one CICADA ligand (all cells in Fig. 4e are CICADA responsive by definition vs Fig. 8e includes non-responders), each cell’s responses had the minimum response to any CICADA ligand subtracted to make the smallest value 0 and then the data was divided by the maximum response of that cell, i.e. taking the 8 data points for amplitudes of response to CICADA, the formula is as follows: $(Cell_{Response} - Cell_{min})/Cell_{max}$. For cells in Fig. 8e which did not respond to any CICADA ligands, normalizing by a very low maximum did not place the data in the correct context so, if a cell did not respond to any CICADA ligand it was instead of being normalized to the cell’s maximum response it was instead divided by the average response to all CICADA ligands so that we place sub-threshold responses within the appropriate context and not emphasizing noise among non-responders. Data were then averaged within each indicated population to generate the % maximum response.”

11. (Response point 37) The initial proposition was to rephrase the legend of [Supplementary Fig. 4] panel i, as the most superficial cells don’t really show a “significant tuning for heat”, but rather show relatively more tuning for heat than in other laminae

This has been rephrased as follows:

“Consistent with prior literature, thermal stimuli have relatively stronger responses in the superficial layers compared with deeper populations”

12. (Manuscript I. 208-210) While Fig. 4i directly presents the degree of polymodality, this does not appear explicitly in suppl. Fig. 6d-g. The sentence might need rephrasing.

We agree, thank you. This has been changed.

“We also found that the tuning of lamina I CICADA populations were distinct from depth matched cells which were not marked by CICADA (Supplementary Fig. 6d-g). This suggests that even when the relative laminar location is controlled for, CICADA marks distinct populations.”

13. (Manuscript I. 310) It should be specified that the manuscript presents data on “excitatory” neurons of the “superficial” dorsal horn.

This has been corrected.

14. (Manuscript I. 315, 327) The word “pain” should be carefully used if referring to animal studies.

This has been clarified.

“We then describe the spinal representations of the intradermal injection of capsaicin, which gives rise to transient burning pain and persistent mechanical allodynia in humans”

& “pain that follows capsaicin treatment in human subjects”

15. (Manuscript I. 322) Subtitle requires rephrasing, authors do not demonstrate that the Ex1 population initiates central sensitization.

This has been corrected.

“Ex1_{TACR3/1} neurons are the initial responders to capsaicin”

16. (Manuscript I. 673) Probably transverse sections?

Yes, transverse. This has been clarified.

17. (Suppl. Fig. 6) Panel e presents cardinal tuning and mentions that they are compared to the AVERAGE of all non-labelled lamina I cells? Doesn't the comparison take into account all of the individual points? The legend also indicates that the vector angle and magnitude are compared with the AVERAGE of ALL lamina I cells (not individual points? Not only of lamina I cells not-labelled with CICADA?)

It should have been referring to cells not marked by CICADA ligands, i.e., unlabeled Lam I cells, this has been corrected.

“Cardinal tuning of lamina I CICADA populations compared to lamina I cells not labeled by CICADA ligands.”